**Enrichment of calcium in sea spray aerosol: Insights from bulk measurements**
**and individual particle analysis during the R/V *Xuelong* cruise in the**
**summertime Ross Sea, Antarctica**
Bojiang Su [a, b], Xinhui Bi [a, c], Zhou Zhang [b, d], Yue Liang [e], Congbo Song [f], Tao Wang [a, b], Yaohao
Hu [a, b], Lei Li [g, *], Zhen Zhou [g], Jinpei Yan [h], Xinming Wang [a, c], Guohua Zhang [a, c, *]
[a] State Key Laboratory of Organic Geochemistry and Guangdong Provincial Key Laboratory of
Environmental Protection and Resources Utilization, Guangzhou Institute of Geochemistry,
Chinese Academy of Sciences, Guangzhou 510640, China
[b] University of Chinese Academy of Sciences, Beijing 100049, China
[c] Guangdong-Hong Kong-Macao Joint Laboratory for Environmental Pollution and Control,
Guangzhou 510640, China
[d] State Key Laboratory of Isotope Geochemistry, Guangzhou Institute of Geochemistry, Chinese
Academy of Sciences, Guangzhou 510640, China
[e] Department of Civil and Environmental Engineering, Faculty of Science and Technology,
University of Macau, Taipa, Macau, China
[f] National Centre for Atmospheric Science (NCAS), University of Manchester, Manchester M13
9PL, UK
[g] Institute of Mass Spectrometry and Atmospheric Environment, Jinan University, Guangzhou
510632, China
[h] Key Laboratory of Global Change and Marine Atmospheric Chemistry, Third Institute of
Oceanography, Ministry of Natural Resources, Xiamen 361005, China
*Corresponding author: zhanggh@gig.ac.cn; lileishdx@163.com

**Abstract:** Although calcium is known to be enriched in sea spray aerosols (SSAs), the factors that affect its enrichment remain ambiguous. In this study, we examine how environmental factors affect the distribution of water-soluble calcium ($Ca^{2+}$) distribution in SSAs. We obtained our dataset from observations taken during a research cruise on the R/V *Xuelong* cruise in the Ross Sea, Antarctica, from December 2017 to February 2018. Our observations showed that the enrichment of $Ca^{2+}$ in aerosol samples was enhanced under specific conditions, including lower temperatures ($< -3.5$ ℃), lower wind speeds ($< 7$ m s$^{-1}$), and the presence of sea ice. Our analysis of individual particle mass spectra revealed that a significant portion of calcium in SSAs was likely bound with organic matter (in the form of a single-particle type, OC-Ca). Our findings suggest that current estimations of $Ca^{2+}$ enrichment based solely on water-soluble $Ca^{2+}$ may be inaccurate. Our study is the first to observe a single-particle type dominated by calcium in the Antarctic atmosphere. Our findings suggest that future Antarctic atmospheric modeling should take into account the environmental behavior of individual OC-Ca. With the ongoing global warming and retreat of sea ice, it is essential to understand the mechanisms of calcium enrichment and the mixing state of individual particles to better comprehend the interactions between aerosols, clouds, and climate during the Antarctic summer.

**Key points:**

- $Ca^{2+}$ enrichment in sea spray aerosols (SSAs) was observed at lower ambient temperatures, lower wind speeds, and in the presence of sea ice.

- Individual particle analysis revealed a significant portion of internally mixed organics with calcium particles in the Antarctic summer atmosphere.

- Current water-soluble estimation of $Ca^{2+}$ enrichment in SSAs may be inaccurate without considering organically complexed calcium.

**Keywords:**

Sea spray aerosol; Calcium enrichment; Individual particle analysis; Environmental factors; Internally mixed organics with calcium particles; Antarctic summer atmosphere.

## 1 Introduction

Sea spray aerosols (SSAs) govern radiative forcing by directly scattering and absorbing solar radiation over the remote ocean (Murphy et al., 1998), and they affect the microphysical properties of marine clouds by serving as cloud condensation nuclei (CCN) and ice nuclei (IN) (Wilson et al., 2015; Brooks and Thornton, 2018; Willis et al., 2018). Calcium is one of the components of SSA, which can present as inorganic calcium (e.g., $CaCl_2$ and $CaSO_4$) (Chi et al., 2015) as well as organic calcium (i.e., $Ca^{2+}$ can readily induce the gelation of organic matter, presenting as the most efficient gelling agent) (Carter-Fenk et al., 2021). Calcium enrichment and chemical signature can affect some physicochemical properties of SSAs such as alkalinity and hygroscopicity (Salter et al., 2016; Mukherjee et al., 2020), which is critical for understanding aerosol-cloud interactions over the remote marine boundary layer (Keene et al., 2007; Leck and Svensson, 2015; Bertram et al., 2018).

Several studies have demonstrated significant enrichment of calcium ($Ca^{2+}$) in SSAs compared to bulk seawater, as briefly summarized in **Table S1** and documented by Keene et al. (2007), Hara et al. (2012), Cochran et al. (2016), Salter et al. (2016), Cravigan et al. (2020), and Mukherjee et al. (2020). For example, Hara et al. (2012) found that the $Ca^{2+}$ enrichment of aerosol samples was sensitive to sea salt fractionation during the cold winter-spring season over the Antarctic coast. Leck and Svensson (2015) suggested that $Ca^{2+}$ enrichment in SSAs is attributed to bubble bursts on sea ice leads over the Arctic area. Similarly, low wind-driven bubble bursts were regarded as a major reason for the $Ca^{2+}$ enrichment in SSAs during an Arctic cruise (Mukherjee et al., 2020). These results shed light on the $Ca^{2+}$ enrichment process; however, our understanding of how environmental factors synergistically affect such enrichment processes remains unclear.

To date, a unified consensus on the chemical form of calcium to explain calcium enrichment
in SSAs has not been reached. Two hypotheses have been proposed: (i) Calcium enrichment is
dominated by inorganic calcium, such as $CaCO_3$ and $CaCl_2$. $Ca^{2+}$ is enriched close to the seawater
surface in the form of ionic clusters (most probably with carbonate ions) (Salter et al., 2016).
Another source of $CaCO_3$ is directly from calcareous shell debris (Keene et al., 2007). Through
bubble bursts, both $CaCO_3$ and $CaCl_2$ along with sea salt can be emitted into the atmosphere. In
addition, the sea salt fractionation by precipitation of ikaite ($CaCO_3 \cdot 6H_2O$) may contribute to
calcium enrichment in aerosol during the freezing of sea ice (Hara et al., 2012). (ii) Calcium
enrichment is attributed to organically complexed calcium. $Ca^{2+}$ may bind with organic matter,
which is relevant with marine microgels and/or coccolithophore phytoplankton scales, and can be
emitted by bubble bursting (Oppo et al., 1999; Sievering, 2004; Leck and Svensson, 2015;
Cochran et al., 2016; Kirpes et al., 2019; Mukherjee et al., 2020). The chemical form of calcium
can determine its atmospheric role. Inorganic calcium may exhibit stronger aerosol alkalinity and
hygroscopicity than organic calcium (Salter et al., 2016; Mukherjee et al., 2020). However, current
estimations of calcium enrichment based solely on water-soluble $Ca^{2+}$ may not precisely explain
the calcium distribution in SSAs. This is because the amount of low water-soluble complexation
of $Ca^{2+}$ with organic matter (e.g., aged $Ca^{2+}$-assembled gel-like particles) (Orellana and Verdugo,
2003; Leck and Bigg, 2010; Russell et al., 2010; Orellana et al., 2011; Leck and Svensson, 2015)
and insoluble $Ca^{2+}$ in the form of calcareous shell debris or the like may not be considered. Thus,
an alternative method, such as discerning the mixing state based on single-particle analysis, may
provide unique insights into the chemical form of calcium, and thus the mechanisms of calcium
enrichment in SSAs.
As a part of the 34[th] Chinese Antarctic Research Expedition (CHINARE ANT34th), this
study aimed to investigate the influencing factors and possible mechanisms of calcium enrichment
in SSAs through R/V *Xuelong* cruise observation campaigns over the Ross Sea, Antarctica. An in-
situ gas and aerosol composition monitoring system (IGAC) was employed to determine the
extent of $Ca^{2+}$ enrichment in SSAs. Single-particle aerosol mass spectrometry (SPAMS) was
utilized to measure the size and chemical signature (i.e., mixing state) of individual calcareous
particles. We first investigated the impact of environmental factors such as ambient temperature,
wind speed, sea ice fraction, chlorophyll-a concentration, and back trajectory coverage on $Ca^{2+}$
enrichment in SSAs. Then, the mechanisms of calcium enrichment in SSAs were inferred
according to the mixing state of individual calcareous particles.
**2 Methodology**
**2.1 The R/V *Xuelong* cruise and observation regions**
Our study focused on the Ross Sea region of Antarctica (50 to 78° S, 160 to 185° E) (**Fig. 1**),
where we conducted two separate observation campaigns aboard the R/V *Xuelong*. During the
observations, this region was relatively isolated from the impact of long-range transport of
anthropogenic aerosols and has experienced the sea ice retreat (Yan et al., 2020a).
The first observation campaign (Leg I) took place from December 2-20, 2017, during the sea
ice period. The second campaign (Leg II) was conducted from January 13 to February 14, 2018,
during the period without sea ice. The sampling design for Leg I and Leg II aimed to investigate
how changing environmental factors affect the enrichment extent of calcium and the
characteristics of individual particles.

**2.2 Meteorological parameters and satellite data of air masses, sea ice, and chlorophyll-a**

We measured various meteorological parameters, such as ambient temperature, relative humidity (RH), wind speed, and true wind direction using an automated meteorological station located on the top deck of the R/V *Xuelong* (**Fig. S1 and Table S2**).

To determine the type of air masses, we first overviewed the 72-hour back trajectory with daily resolution per each starting location by using the NOAA Hybrid Single-Particle Lagrangian Integrated Trajectories (HYSPLIT, version 4.9) model (**Fig. S2**). Additionally, we conducted a 96-hour back trajectory analysis with an hourly resolution, which covered the enhanced calcium enrichment events associated with sea ice fraction and chlorophyll-a concentration (discussed in section 3.1), using the TrajStat in Meteoinfo (version 3.5.8) (Wang et al., 2009; Wang, 2014). Meteorological data used for back trajectory analysis obtained from the Global Data Assimilation System (GDAS, ftp://ftp.arl.noaa.gov/pub/archives). Moreover, we obtained the monthly sea ice fraction from the Sea Ice Concentration Climate Data Record with a spatial resolution of 25 km (https://www.ncei.noaa.gov/products/climate-data-records/sea-ice-concentration) and the 8-day chlorophyll-a concentration from MODIS-aqua with a spatial resolution of 4 km (https://modis.gsfc.nasa.gov) (**Fig. S3**).

During the R/V *Xuelong* cruise observation campaigns, leg I was dominantly affected by the air masses from the sea ice-covered open water (92%, by trajectory coverage), and leg II was mainly affected by the air masses from continental Antarctica (58%) (**Table S2**). The average ambient temperature (-4.0 ± 1.4 ℃ vs. -3.1 ± 2.2 ℃), wind speed (7.2 ± 5.5 m s$^{-1}$ vs. 7.1 ± 4.2 m s$^{-1}$), and chlorophyll-a concentration (0.51 ± 0.29 μg L$^{-1}$ vs. 0.44 ± 0.18 μg L$^{-1}$) varied slightly between the legs I and II (**Table S2**).

**2.3 Contamination control during observation campaigns**
During the research cruise, the major contamination source was identified as emissions from
a chimney located at the stern of the vessel and about 25 m above the sea surface. To mitigate the
potential impact of ship emissions on aerosol sampling, we have taken several measures. Firstly, a
total suspended particulate (TSP) sampling inlet connecting to the monitoring instruments was
fixed to a mast 20 m above the sea surface, located at the bow of the vessel. In addition, the
sampling inlet was fixed on a ship pillar with a rain cover, which could minimize the potential
influence of violent shaking of the ship and sea waves. Secondly, sampling was only conducted
while the ship was sailing, to avoid the possible effect of ship emission on aerosol sampling under
the low diffusion condition. Lastly, we did not observe the mass spectral characteristics associated
with ship emission (e.g., particles simultaneously contain $m/z$ 51 $[V]^+$, 67 $[VO]^+$, and element
carbon) during the observation campaigns (Liu et al., 2017; Passig et al., 2021). These measures
ensured that the collected data were representative and reliable for subsequent analysis.
**2.4 Instrumentation**
An IGAC (Model S-611, Machine Shop, Fortelice International Co. Ltd.) and a SPAMS
(Hexin Analytical Instrument Co., Ltd.) were synchronously employed to determine water-soluble
ion mass concentrations of bulk aerosol and the size and chemical composition of individual
particles in real-time with hourly resolution (**Figs. 2** and **S4**). In the aerosol sampling procedure, a
TSP inlet with a $PM_{10}$ cyclone (trap efficiency greater than 99% for particles > 0.3 μm, $D_{a50}$ = 10
± 0.5 μm) was used for IGAC sampling and a $PM_{2.5}$ cyclone ($D_{a50}$ = 2.5 ± 0.2 μm) to remove
particles larger than 2.5 μm for SPAMS. All instruments were connected using conductive silicon
tubing with an inner diameter of 1.0 cm.

**2.4.1 Aerosol water-soluble ion constituents**

The details of the analytical method of IGAC have been described in previous studies (Young
et al., 2016; Yan et al., 2019; Yan et al., 2020b). Briefly, the IGAC system consisted of three main
units, including a Wet Annular Denuder (WAD), a Scrub and Impact Aerosol Collector (SIAC),
and an ion chromatograph (IC, Dionex ICS-3000) (**Fig. 2**). Gases and aerosols were passed
through WAD with a sampling flow of 16.7 L min$^{-1}$. Two concentric Pyrex glass cylinders with a
length of 50 cm and inner and outer diameters of 1.8 and 2.44 cm were assembled to WAD, in
which the inner walls of the annulus were wetted with ultrapure water (18.2 MΩ cm$^{-1}$). This part
was responsible for the collection of acidic and basic gases by diffusion and absorption of a
downward-flowing aqueous solution. The SIAC had a length of 23 cm and a diameter of 4.75 cm,
which was positioned at an angle to facilitate the collection of enlarged particles. The collected
particles were separated firstly, continually enlarged by vapor steam, and then accelerated through
a conical-shaped impaction nozzle and collected on an impaction plate. Each aerosol sample was
collected for 55 minutes and injected for 5 minutes. The injection loop size was 500 μL for both
anions and cations, which were subsequently analyzed by IC. The collection efficiency of aerosol
and gas samples before they entered IC was previously reported higher than 89% (for 0.056 μm
particles, 89%; for 1 μm particles, 98%; for gas samples, > 90%) (Chang et al., 2007; Tian et al.,
2017). The target ion concentrations were calibrated with a coefficient of determination (r$^2$) above
0.99 by using standard solutions (0.1-2000 μg L$^{-1}$). The detection limits for Na, Cl, Ca, K, and Mg
were 0.03, 0.03, 0.019, 0.011, and 0.042 μg L$^{-1}$ (aqueous solution), respectively. The systematic
error of the IC systems was generally less than 5%. The detection limits for $Na^+$, $Cl^-$, $Ca^{2+}$, $K^+$, and
$Mg^{2+}$ were 0.03, 0.03, 0.019, 0.011, and 0.042 $\mu g\ L^{-1}$ (aqueous solution), respectively.
Throughout the observation campaigns, the mean $Na^+$ and $Ca^{2+}$ mass concentrations were
364.64 ng $m^{-3}$ (ranging from 6.66 to 4580.10 ng $m^{-3}$) and 21.20 ng $m^{-3}$ (ranging from 0.27 to
334.40 ng $m^{-3}$), respectively, which were 10 times higher than the detection limits. Analytical
uncertainty of $Ca^{2+}$ enrichment based on water-soluble analysis was estimated at less than 11%
(**Supporting Information, SI text S1**).
It should be clarified that the water-soluble ion mass concentration included the pure
inorganic part (e.g., pure sea salt, NaCl) and mixed organic-inorganic part (e.g., gel-like particles)
(Quinn et al., 2015). Numerous studies have reported that primary SSAs exhibited moderate
hygroscopicity and water solubility due to a certain water-soluble organic fraction (~ 25%, by
mass), such as carboxylates, lipopolysaccharides (LPSs), humic substances, and galactose (Oppo
et al., 1999; Quinn et al., 2015; Schill et al., 2015; Cochran et al., 2017). For example, Oppo et al.
(1999) indicated that humic substances were an important pool of water-soluble natural surfactants
(40-60%) in marine surfactant organic matter. In addition, LPSs are preferentially transferred to
submicron SSAs during bubble bursting and exhibit a certain solubility of 5 g $L^{-1}$ in pure water.
(Facchini et al., 2008; Schill et al., 2015). Therefore, both organic and inorganic parts with a
water-soluble nature could be retained, contributing to the water-soluble ion mass concentration
(e.g., $Ca^{2+}$).
**2.4.2 Single-particle analysis**
A brief description of SPAMS has been provided elsewhere (Li et al., 2011). Briefly, the

aerosols were drawn into SPAMS by a PM$_{2.5}$ inlet after a silica gel dryer (**Fig. 2**). A collimated

particle beam focused by an aerodynamic lens was then accelerated in an accelerating electric

field and passed through two continuous laser beams (Nd: YAG laser, 532 nm). The obtained time

of flight (TOF) and velocity of individual particles were used to calculate the vacuum

aerodynamic diameter ($D_{va}$) based on a calibration curve. Subsequently, particles with a specific

velocity were desorbed and ionized by triggering a pulse laser (an Nd: YAG laser, 266 nm, 0.6 ±

0.06 mJ was used in this study). The ion fragments were recorded using a bi-polar TOF mass

spectrometer. The detectable dynamic mass spectral ion signal is 5-20,000 mV. Before the use of

SPAMS, standard polystyrene latex spheres (0.2-2 μm, Duke Scientific Corp.) and PbCl$_2$ and

NaNO$_3$ (0.35 μm, Sigma-Aldrich) solutions were used for the size and mass spectral calibration,

respectively. The hit rate, defined as the ratio of ionized particles to all sampled particles, of the

SPAMS was ~ 11% during the cruise observation campaigns.

During the R/V *Xuelong* cruise observation campaigns, approximately 930,000 particles with

mass spectral fingerprints and $D_{va}$ ranging from 0.2 to 2 μm were measured. An adaptive

resonance theory neural network (ART-2a) was used to group the particles into several clusters

based on their mass spectral fingerprints, using parameters of a vigilance factor of 0.85, a learning

rate of 0.05, and a maximum of 20 iterations (Song and Hopke, 1999). The manually obtained

clusters were sea salt (SS, 16.5%), aged sea salt (SS-aged, 8.1%), sea salt with biogenic organic

matter (SS-Bio, 3.1%), internally mixed organics with calcium (OC-Ca, 48.7%), internally mixed

organics with potassium (OC-K, 13.7%), organic-carbon-dominated (OC, 7.0%), and element

carbon (EC, 2.9%) (**Fig. S5 and Table S3**) (Prather et al., 2013; Collins et al., 2014; Su et al.,

2021). All single-particle types had marine origins with typical mass spectral characteristics of Na

(*m/z* 23), Mg (*m/z* 24), K (*m/z* 39), Ca (*m/z* 40), and Cl (*m/z* -35 and -37), except for EC (**SI text**
**S2**). There was little difference in individual particle analysis regarding chemical composition,
size, and mixing state of particle clusters obtained from leg I and leg II (**SI Text S3**).
**3 Results**
**3.1 $Ca^{2+}$ enrichment dominated by environmental factors**
We propose that both $Na^+$ and $Ca^{2+}$ in our observations originated from marine sources.
The mass concentration of $Na^+$ exhibited a strong positive correlation with that of $Cl^-$ (r = 0.99,
$p < 0.001$) and $Mg^{2+}$ (r = 0.99, $p < 0.001$) (**Fig. S6**), indicating that they had a common origin
(i.e., sea spray). However, it is not surprising that the mass concentration of $Na^+$ showed a
relatively weak correlation with that of $Ca^{2+}$ (r = 0.51, $p < 0.001$) (**Fig. S6**). This can be
explained by the low water-soluble complexation of $Ca^{2+}$ with organic matter and/or insoluble
$Ca^{2+}$ in the form of calcareous shell debris, such as $CaCO_3$. In addition, the potential impact of
long-range transport of anthropogenic aerosols and dust contributing to $Ca^{2+}$ may be limited due
to the predominance of polar air masses during the observation campaigns (**Fig. S2**).
The enrichment factor (EF*x*), defined as the mass concentration ratio of a specific species
*X* to $Na^+$ in aerosols to that in bulk seawater, is generally used to describe the enrichment extent
of species *X* in aerosols.
$$\mathrm{EF}x = \frac{([X]/[Na^+])_{aerosol}}{([X]/[Na^+])_{seawater}}$$
An EF*x* > 1 indicates a positive enrichment; otherwise, it indicates depletion. Generally, the
ratio of $Ca^{2+}$ to $Na^+$ in seawater is 0.038 (w/w) (Boreddy and Kawamura, 2015; Su et al., 2022).
During the whole cruise, the hourly average $EF_{Ca}$ was 2.76 ± 6.27 (mean ± standard deviation (M

± SD), n = 1051, ranged from 0.01 to 85, median =1.18, interquartile range (IQR) = 1.85). Similar

to previous studies (Salter et al., 2016), positive magnesium ($Mg^{2+}$) and potassium ($K^+$)

enrichment in SSAs was also observed (**SI text S4**).

   **Figure 3** presents the enrichment factor of $Ca^{2+}$ ($EF_{Ca}$) at different ambient temperatures

(separated by a mean value of -3.5 °C), wind speeds (separated by a mean value of 7 m s$^{-1}$), and in

the presence/absence of sea ice during the entire observation campaign. The results indicated that

the highest $EF_{Ca}$ zone (M ± SD = 3.83 ± 3.43, median = 2.66, IQR = 3.37, n = 144) occurred at a

lower ambient temperature (< -3.5 °C), lower wind speed (< 7 m s$^{-1}$) and in the presence of sea ice

(**Fig. 3d**). Compared to the contrary conditions (i.e., ambient temperatures ≥ -3.5 °C, wind speeds

≥ 7 m s$^{-1}$, and the absence of sea ice), there was almost calcium depletion ($EF_{Ca}$, M ± SD = 1.01 ±

0.80, median = 0.70, IQR = 0.73, n = 182) (**Fig. 3c**). Notably, we observed a higher $EF_{Ca}$ during

the sea ice period than during the period without sea ice (3.83 ± 3.43 vs. 2.45 ± 3.09 by M ± SD

and 2.66 vs. 1.18, by median) (**Fig. 3d**), under the conditions of ambient temperatures < -3.5 °C

and wind speeds < 7 m s$^{-1}$. In addition, we also observed more frequent $Ca^{2+}$ enrichment events

during the sea ice period (71.0% in leg I) compared to the period without sea ice (47.7% in leg II)

(**Table S2**). Moreover, the increased $EF_{Ca}$ varied with decreasing ambient temperature and wind

speed and with increasing sea ice fraction, as shown in **Fig. 4**. Taken together, our results indicate

that the enhanced $Ca^{2+}$ enrichment in SSAs is sensitive to the lower temperature, lower wind

speeds, and the presence of sea ice.

   We further analyzed the distribution of $Ca^{2+}$ enrichment concerning 96-hour back trajectories

with sea ice fraction and chlorophyll-a concentration, as shown in **Fig. 5**. During the observation

campaigns, we identified five areas with continuous enhancement of $Ca^{2+}$ enrichment, namely,

Area 1 and 2 during the leg II, and Area 3,4, and 5 during the leg I. Our results indicated that air
masses traveling over the sea ice and marginal ice zone (> 95%, by trajectory coverage) in Areas 3,
4, and 5, as well as those over the sea ice (28%-33%) and land-based Antarctic ice (57-59%) in
Area 1 and 2, were strongly associated with the increased calcium enrichment (**Table S4**). These
pieces of evidence further support the influence of sea ice on the increased calcium enrichment,
while simultaneously ruling out the influence of long-range transport of anthropogenic aerosol and
dust outside the Antarctic.

We observed that a series of high $EF_{Ca}$ cases in Area 1 were associated with a high

concentration of chlorophyll-a ($0.99 \pm 1.65$ µg L$^{-1}$). However, it is unlikely that phytoplankton
and/or bacteria are responsible for the enhanced $EF_{Ca}$ cases due to the weak correlation (r = 0.12, *p*
< 0.01) between the chlorophyll-a concentration and $EF_{Ca}$ values (**Fig. S7**). Moreover, although
the ship track of leg II covered large areas with high chlorophyll-a concentrations, the high $EF_{Ca}$
values were only present at the narrow temporal and spatial scales. Furthermore, results from back
trajectories indicated that air masses did not significantly travel through the region with elevated
chlorophyll-a concentration. Therefore, we suggest that the impact of chlorophyll-a concentration
on $Ca^{2+}$ enrichment may be limited.
**3.2 Single-particle characteristics of Ca-containing particles**

To elucidate the mixing state of individual calcareous particles, we set a threshold for the ion

count rate of *m/z* 40 [Ca]$^{+}$ (ion intensity > 100 mV) to reclassify all single-particle types that were
obtained from the ART-2a algorithm. This means that all reclassified particles contain signals of
*m/z* 40 [Ca]$^{+}$. A total of ~ 580, 000 Ca-containing particles were distributed among all particle

types, accounting for ~ 62% of the total obtained particles. OC-Ca was the dominant (~ 72%, by occurrence frequency) particle type among all Ca-containing particles, followed by SS-Ca (calcium-containing sea salt, ~ 12%) (**Fig. 6h**). Each of the remaining particle types accounted for negligible fractions (< 7%) in the total of Ca-containing particles, and were classified as "Other". Thus, they were not included in the following discussion.

OC-Ca was characterized by a prominent ion signature for $m/z$ at 40 $[Ca]^+$ in the positive mass spectrum and organic marker ions of biological origin (e.g., organic nitrogen, phosphate, carbohydrate, siliceous materials, and organic carbon) in the negative spectrum (**Fig. 6d**). Specifically, organic nitrogen ($m/z$ -26 $[CN]^-$ and -42 $[CNO]^-$) showed the largest number fraction (NF) at ~88% (**Fig. S5h**), which is likely derived from organic nitrogen species, such as amines amino groups, and/or cellulose (Czerwieniec et al., 2005; Srivastava et al., 2005; Köllner et al., 2017; Dall'osto et al., 2019). Higher NFs of phosphate (16%; $m/z$ -63 $[PO_2]^-$ and -79 $[PO_3]^-$), carbohydrates (24%; $m/z$ -45 $[CHO_2]^-$, -59 $[C_2H_3O_2]^-$, and -73 $[C_3H_5O_2]^-$), siliceous materials (40%; $m/z$ -60 $[SiO_2]^-$), and organic carbon (37%; $m/z$ 27 $[C_2H_3]^-$ and 43 $[C_2H_3O_3]^-$) were also observed in OC-Ca relative to other particle types (**Fig. S5h**). These organic ion signatures likely correspond to phospholipids, mono- and polysaccharides, and biosilica structures (e.g., exoskeletons or frustules), which may be derived from the intact heterotrophic cells, fragments of cells, and exudates of phytoplankton and/or bacterial (Prather et al., 2013; Guasco et al., 2014; Zhang et al., 2018). Besides, the strong organic ion intensities may truly reflect the amount of organic material in OC-Ca, because the particles are sufficiently dry during the ionization process (i.e., complete positive and negative mass spectra) (Gross et al., 2000). Notably, the possible ion signals of bromide ($m/z$ -79 and -81) were observed in OC-Ca, indicating a potential source of

blowing snow (Yang et al., 2008; Song et al., 2022).
The OC-Ca particles are most likely classified as a distinct SSA population, probably of
marine biogenic origin. Sea salt particles typically exhibit a stronger $m/z$ 23 $[Na]^+$ than $m/z$ 40
$[Ca]^+$ due to the higher concentration of $Na^+$ vs. $Ca^{2+}$ in seawater and also due to the lower
ionization potential of Na vs. Ca (5.14 eV vs. 6.11 eV) (Gross et al., 2000). However, the ratio of
$m/z$ 23 $[Na]^+$ to $m/z$ 40 $[Ca]^+$ in the OC-Ca particles is reversed, verifying a distinct single particle
type (Gross et al., 2000; Gaston et al., 2011). Similarly, the ion signal of $m/z$ 39 $[K]^+$ does not
surpass that of $m/z$ 40 $[Ca]^+$ in OC-Ca, although K is ionized more easily than Ca (4.34 eV vs.
6.11eV) (Gross et al., 2000). Although RH at the sampling outlet was < 40%, the short residence
time of the particles within the drying tube (< 5 s) and vacuum system (< 1 ms) could have been
insufficient for the complete efflorescence of SSAs (Gaston et al., 2011; Sierau et al., 2014).
Hence, the OC-Ca could not be attributed to the chemical fractionation of the efflorescence SSAs
in SPAMS analysis. Additionally, based on the single-particle mass spectrometry technique, some
particle types with similar chemical characteristics to OC-Ca have been observed in both field and
laboratory studies (e.g., atomization of sea ice meltwater collected in the Southern Ocean) (Gaston
et al., 2011; Prather et al., 2013; Collins et al., 2014; Guasco et al., 2014; Dall'osto et al., 2019; Su
et al., 2021). The OC-Ca may be from local emissions because the measurements were almost
entirely influenced by polar air masses (**Fig. S1**). Other possible sources, such as glacial dust
(Tobo et al., 2019), could be excluded because of the lack of crustal mass spectral characteristics
(e.g., -76 $[SiO_3]^-$, 27 $[Al]^+$, and 48 $[Ti]^+$/64 $[TiO]^+$) (Pratt et al., 2009; Zawadowicz et al., 2017).
And the mean mass concentration ratio of Ca/Na in the aerosol sample was only 0.10, much lower
than that in the crust (1.78, w/w).

In contrast, SS-Ca was classified as a pure inorganic cluster with predominant contributions

of Na-related compounds (*m/z* 23 $[Na]^+$, 46 $[Na_2]^+$, 81/83 $[Na_2{}^{35/37}Cl]^+$, and -93/-95 $[Na^{35/37}Cl_2]^-$),
Mg (*m/z* 24), K (*m/z* 39), and Ca (*m/z* 40) in the mass spectra (**Fig. 6a**). Organic ion signals such
as organic nitrogen (*m/z* -26 $[CN]^-$ and -42 $[CNO]^-$) and phosphate (*m/z* -63 $[PO_2]^-$ and -79 $[PO_3]^-$)
were rarely detected (~1%, by NF). As described above, these compounds relate to oceanic
biogeochemical processes. In addition, secondary species (e.g., nitrate of *m/z* -62 $[NO_3]^-$ and
sulfate of *m/z* -97 $[HSO_4]^-$) were also not observed, indicating a fresh origin and/or less
atmospheric aging. As a subpopulation of SS, SS-Ca may originate from bubble bursting within
open water and/or blowing snow.
**4 Discussion**

SS-Ca (calcium-containing sea salt) represents a mixture of NaCl and $CaCl_2$. However, the

SS-Ca showed a weak correlation (r = 0.21, *p* < 0.05, by count and r = 0.03, *p* < 0.05, by the peak
area of *m/z* 40 $[Ca]^+$) with the mass concentration of $Ca^{2+}$ (**Table 1**). In addition, the proportion of
SS-Ca was also small (11.6%, **Fig. 6h**). These results indicate that $CaCl_2$ is not the major reason
for the $Ca^{2+}$ enrichment in SSAs, although $CaCl_2$ has been proposed as a cause, based on
laboratory atomizing of pure inorganic artificial seawater (Salter et al., 2016). The contribution of
ikaite ($CaCO_3 \cdot 6H_2O$) could also be excluded due to its low water solubility (Bischoff et al., 1993;
Dieckmann et al., 2008; Dieckmann et al., 2010), although ikaite from sea salt fractionation has
also been proposed to account for the $Ca^{2+}$ enrichment in SSAs over the Antarctic coast (Hara et
al., 2012). Moreover, the mass spectral signatures of $CaCO_3$ (e.g., *m/z* 56 $[CaO]^+$ and -60 $[CO_3]^{2-}$
(see Sullivan et al. (2009)) were also rare in the SS-Ca particles (**Fig. 6a**).

As a major component ($\sim$ 72%, by occurrence frequency) of the Ca-containing particles, OC-

Ca is expected to be partially responsible for the calcium enrichment in SSAs. First, the OC-Ca
and mass concentration of $Ca^{2+}$ exhibited moderately weak positive correlations ($r = 0.42$, $p < 0.05$,
by count and $r = 0.49$, $p < 0.05$, by the peak area of $m/z$ 40 $[Ca]^+$) and moderately strong
correlations under higher $EF_{Ca}$ values ($EF_{Ca} > 10$, $r = 0.63$, $p < 0.05$, by count and $r = 0.68$, $p <$
0.05, by the peak area of $m/z$ 40 $[Ca]^+$) (**Table 1**). Also, such correlations were great during leg I ($r$
$= 0.59$, $p < 0.05$, by count and $r = 0.60$, $p < 0.05$, by the peak area of $m/z$ 40 $[Ca]^+$). Second, the
OC-Ca showed a size distribution with a peak at 1 μm (**Fig. 6i**), which is consistent with the
significant $Ca^{2+}$ enrichment that is generally found in submicron SSAs (Cochran et al., 2016;
Salter et al., 2016; Mukherjee et al., 2020).

We further show that calcium may strongly mix with organic matter, probably as organically

complexed calcium, in the OC-Ca particles. The calcium correlated well with different kinds of
organic matter (e.g., phosphate, $r = 0.81$, $p < 0.05$, by the peak area), but poorly correlated with
chloride ($r = 0.21$, $p < 0.05$, by the peak area and $r = 0.48$, $p < 0.05$, by mass concentration) (**Fig.**
**S6**). In addition, different kinds of organic matter (e.g., organic nitrogen, organic carbon, etc.) in
the OC-Ca particles also showed enrichment trends below the submicron level, analogously to
$Ca^{2+}$ enrichment (**Fig. S8**). Particularly, $EF_{Ca}$ and organic nitrogen (with the largest NF in OC-Ca)
were both affected by the environmental factors of ambient temperature, wind speed, and sea ice
fraction, indicating possible organic binding with calcium (**Fig. S9**).

To exclude the potential inorganic water-soluble compounds (i.e., chloride ($m/z$ -35 and -37),

nitrate ($m/z$ -62), and sulfate ($m/z$ -97)), we further classified OC-Ca into two subpopulations, OC-
Ca-Organic (23.6%, by proportion) and OC-Ca-Inorganic (48.7%, by proportion) (**Fig. S10**),
depending on the presence of inorganic ion signals (i.e., chloride of $m/z$ -35/-37 $[Cl]^-$, nitrate of
$m/z$ -62 $[NO_3]^-$, and sulfate of $m/z$ -97 $[HSO_4]^-$). Both the OC-Ca types and mass concentrations of
$Ca^{2+}$ showed enhanced correlations under high $EF_{Ca}$ values (**Table 1**). In particular, OC-Ca-
Organic exhibited stronger correlations than did OC-Ca-Inorganic (r = 0.51 vs. r = 0.28, $p < 0.05$,
by count and r = 0.51 vs. 0.31, $p < 0.05$, by the peak area of $m/z$ 40 $[Ca]^+$, respectively), which
indicates the importance of OC-Ca-Organic for the enrichment of $Ca^{2+}$. Although we did not
measure the hygroscopicity of the OC-Ca in this study, we infer it to be hygroscopic to some
extent. As reported by Cochran et al. (2017), the mixture of sea salt with organic matter can also
exhibit a certain hygroscopicity (hygroscopicity parameter, 0.50-1.27). Therefore, it is likely that
the organically complexed calcium is slightly water-soluble and is partially responsible for
calcium enrichment, while current studies may neglect it.

The possible processes contributing to the calcium enrichment induced by OC-Ca can only

be speculated on (**Fig. 7**). $Ca^{2+}$ tends to bind with organic matter of biogenic origin, such as
exopolymer substances (EPSs), and subsequently assemble as marine microgels (Verdugo et al.,
2004; Gaston et al., 2011; Krembs et al., 2011; Orellana et al., 2011; Verdugo, 2012; Orellana et
al., 2021). Large amounts of microgels, driven by sea ice algae, microorganisms, and/or exchanges
of organic matter with the seawater below, stick to the sea ice due to its porous nature.
Furthermore, they are likely to be present in the snow, frost flowers, and brine channels (Krembs
et al., 2002; Gao et al., 2012; Vancoppenolle et al., 2013; Arrigo, 2014; Boetius et al., 2015;
Kirpes et al., 2019). A low wind speed may not only be conducive to the formation of frost flowers
and snow but also produce less sea salt (i.e., small yields of $Na^+$ relative to $Ca^{2+}$) (Rankin et al.,
2002). Correspondingly, a high wind speed ($\geq 7$ m s$^{-1}$) can yield more sea salt by blowing-snow
events and/or wave breaking (Yang et al., 2008; Song et al., 2022), presenting a dilution effect of
$Na^+$ on $Ca^{2+}$. In this case, the calcium enrichment in SSAs could reasonably be attributed to the
possible gel-like calcium-containing particles released by low-wind-blown sea ice. This inference
is supported by the observation of air masses blown over a large fraction of sea ice/ land-based
Antarctic ice, as well as a moderate negative correlation (r = 0.50, $p < 0.001$) between wind speed
and sea ice fraction. In addition, we also observed a higher proportion of OC-Ca at low wind
speeds (< 7 m s$^{-1}$, 61.5%) than at high wind speeds (≥ 7 m s$^{-1}$, 38.5%). Coincidently, Song et al.
(2022) also reported that a low wind-blown sea ice process can drive the biogenic aerosol
response in the high Arctic. In addition, the enhanced presence of film drops was observed at
lower wind speeds (< 6 m s$^{-1}$) (Norris et al., 2011), which suggests that the bubble bursts within
the sea ice leads and open water may also be responsible for the release of OC-Ca and its calcium
enrichment involved (Leck and Bigg, 2005b, a; Bigg and Leck, 2008; Leck and Bigg, 2010; Leck
et al., 2013; Kirpes et al., 2019).
As expected, the results of the $Ca^{2+}$ enrichment in SSAs obtained from ion mass
concentration via IGAC did not fully align with results from SPAMS datasets. We propose two
possible explanations for this discrepancy: (i) It could be attributed to a difference in the size of
particles collected by the two different instruments (~ 10 μm for IGAC and 0.2–2 μm for SPAMS).
In addition, SPAMS cannot measure the Aitken-mode particles (Sierau et al., 2014), and can
measure only the tail of accumulation-mode particles with a relatively low hit rate (~11% in this
study). (ii) The types of datasets obtained via IGAC (ion mass concentration) and SPAMS (mass
spectral characteristics) are different. The former method partially reflects the $Ca^{2+}$ distribution
based on water-soluble $Ca^{2+}$, while the OC-Ca measured by SPAMS may have low water

solubility. The latter method is still challenging to use for quantitative measurements due to potential inhomogeneities in the transmission efficiencies of the aerodynamic lenses and desorption/ionization, as well as the matrix effects of individual particles (Gross et al., 2000; Qin et al., 2006; Pratt and Prather, 2012). Therefore, it may not be straightforward to compare the particle count and peak area with the absolute mass concentration.

Although there is a discrepancy between the two instruments, we believe our results to be reliable and representative. On the one side, the quantitative results concluded by IGAC confirm the enrichment of $Ca^{2+}$ in SSAs and demonstrate their dependence on and relevance to the environmental factors. On the other side, the individual particle analysis ranging in size from 0.2 to 2 μm is highly appropriate for revealing the calcium distribution in SSAs, as previous studies have shown increasing $Ca^{2+}$ enrichment in SSAs below 1 μm (Oppo et al., 1999; Hara et al., 2012; Cochran et al., 2016; Salter et al., 2016; Mukherjee et al., 2020). Our study successfully identifies a unique calcareous particle type (i.e., OC-Ca) and its specific mixing state. A comprehensive understanding of the characteristics of OC-Ca to the mechanisms of calcium enrichment is essential for further recognizing the CCN and IN activation in remote marine areas.

Another limitation is that only several environmental factors were considered for calcium enrichment in this study. Some potential factors, such as surface net solar radiation, snowfall, total cloud cover, surface pressure, total precipitation, boundary layer height, seawater salinity, etc., may also affect the calcium enrichment in SSAs through regulating the yield of sea salt (i.e., $Na^+$ mass concentration) (Song et al., 2022). However, they were not available in this study because of the lack of measurement during the cruise. Meanwhile, the satellite data with low temporal-spatial resolution cannot match per hour in each starting condition. We hope that future research will

further investigate the enrichment of specific species in SSAs under a wider range of
meteorological or oceanographic conditions.

**5 Conclusions and atmospheric implications**

We investigated the distribution of calcium in SSAs through the R/V *Xuelong* cruise
observation campaigns over the Ross Sea, Antarctica. The most significant $Ca^{2+}$ enrichment in
SSAs occurred under relatively lower ambient temperatures (< -3.5 °C) and wind speeds (< 7 m s$^{-1}$)
and with the presence of sea ice. With the help of individual particle mass spectral analysis, we
first propose that a single-particle type of OC-Ca (internally mixed organics with calcium),
probably resulting from the preferential binding of $Ca^{2+}$ with organic matter, could partially
account for the calcium enrichment in SSAs. We speculate that OC-Ca is likely produced from the
effects of low wind-blown sea ice on microgels induced by $Ca^{2+}$ and/or the bubble bursts in the
open-water and/or sea ice leads. However, the impact of environmental factors and OC-Ca on
calcium enrichment in SSAs still cannot be well predicted by multiple linear regression and
random forest analysis (**SI text S5**), which may be ascribed to other unknown mechanisms and/or
organically complexed calcium with low water solubility. In addition, our conclusions based on
limited spatial, temporal, meteorological, and oceanographic conditions may not be accessible to
other seasons and oceanic basins.
We suggest that the environmental behaviors of the possible gel-like calcium-containing
particles (i.e., OC-Ca) should be paid more attention behind the mechanisms of calcium
enrichment. Under the stimulation of specific environmental factors (e.g., pH, temperature,
chemical compounds, pollutants, and UV radiation), their physicochemical properties would be
changed (e.g., water-solubility enhanced by the cleavage of polymers) (Orellana and Verdugo,
2003; Orellana et al., 2011). Such particles may be preferred candidates for CCN and/or IN (Willis
et al., 2018; Lawler et al., 2021). To our knowledge, this is the first report of a calcium-dominated
single-particle type OC-Ca in the Antarctic. In the context of global warming and sea ice retreat,
this work provides insight into the chemical composition and distribution of submicron SSAs in
the Antarctic summer atmosphere, which would be helpful for a better understanding of aerosol-
cloud-climate interactions.
**Data Availability Statement**
The data are available at Zenodo (https://doi.org/10.5281/zenodo.8279334). Details can be
accessed by contacting the corresponding author Guohua Zhang (zhanggh@gig.ac.cn) and the first
author Bojiang Su (subojiang21@mails.ucas.ac.cn).
**Declaration of Competing Interest**
The authors declare that they have no known competing financial interests or personal
relationships that could have appeared to influence the work reported in this paper.
**Author Contributions**
The idea for the study was conceived by BJS. BJS analyzed the data, prepared the figures, and
wrote the manuscript under the guidance of GHZ and XYB. LL and JPY contributed to the
observation data. All co-authors contributed to the discussions of the results and refinement of the
manuscript.
**Acknowledgment**
This work was supported by the National Natural Science Foundation of China (42222705 and
42377097), the Youth Innovation Promotion Association CAS (2021354), the Guangdong Basic
and Applied Basic Research Foundation (2019B151502022), and the Guangdong Foundation for
Program of Science and Technology Research (2020B1212060053). We appreciate the Chinese
Arctic and Antarctic Administration for its support in fieldwork. The authors would like to thank
the editor and reviewers for their valuable time and feedback.
**Figure captions**

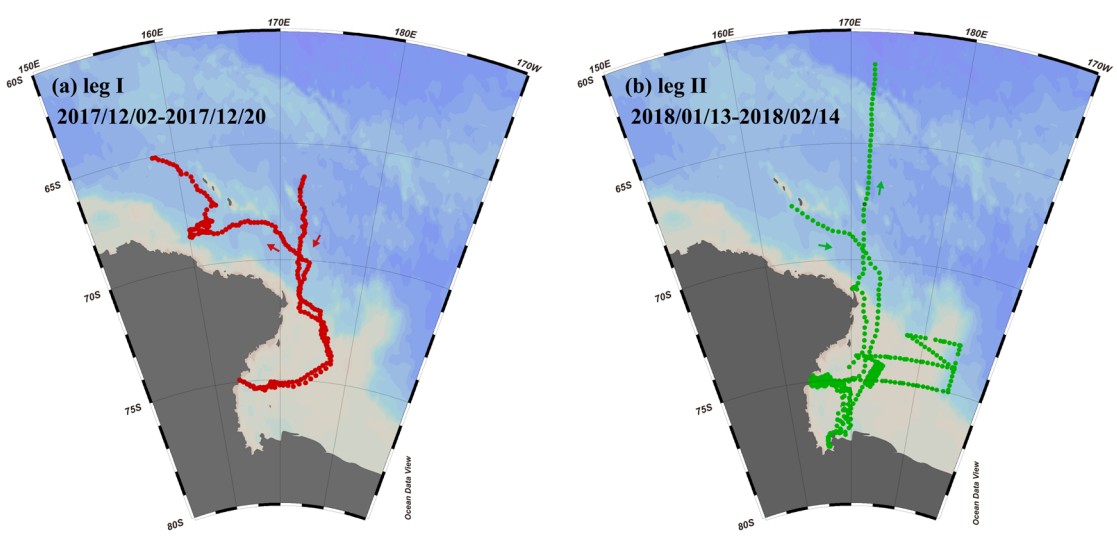


**Figure 1**
Observation campaigns through R/V *Xuelong* in the Ross Sea, Antarctic. (a) Leg I took place from
December 2-20, 2017. (b) Leg II was conducted from January 13 to February 14, 2018.

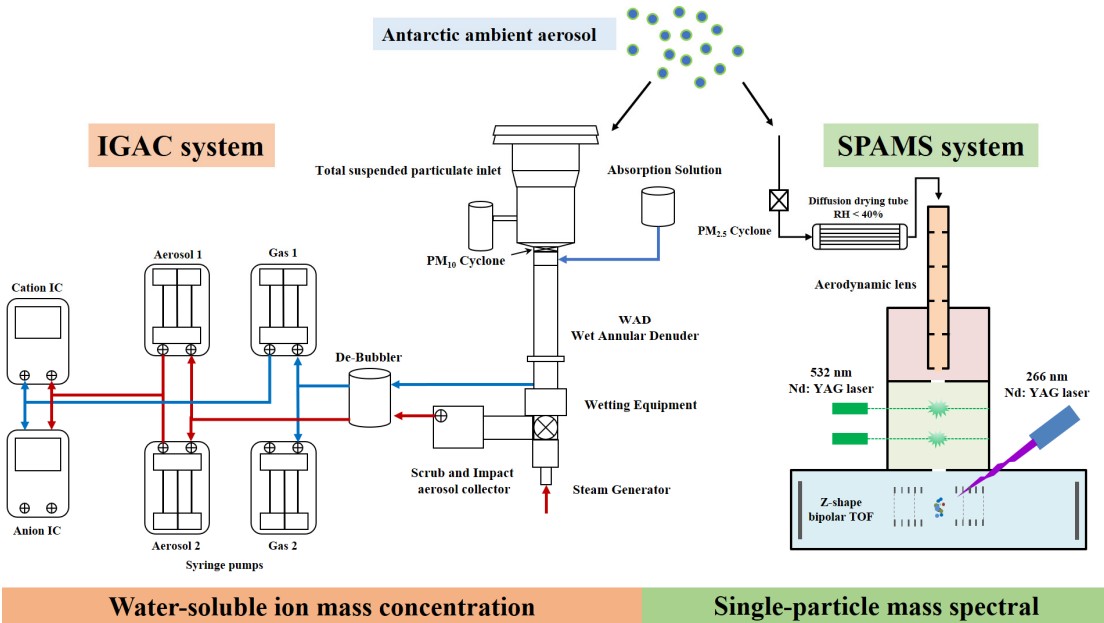


**Figure 2**
A schematic of the aerosol sampling system of IGAC and SPAMS during the research cruise over
the Ross Sea, Antarctic.


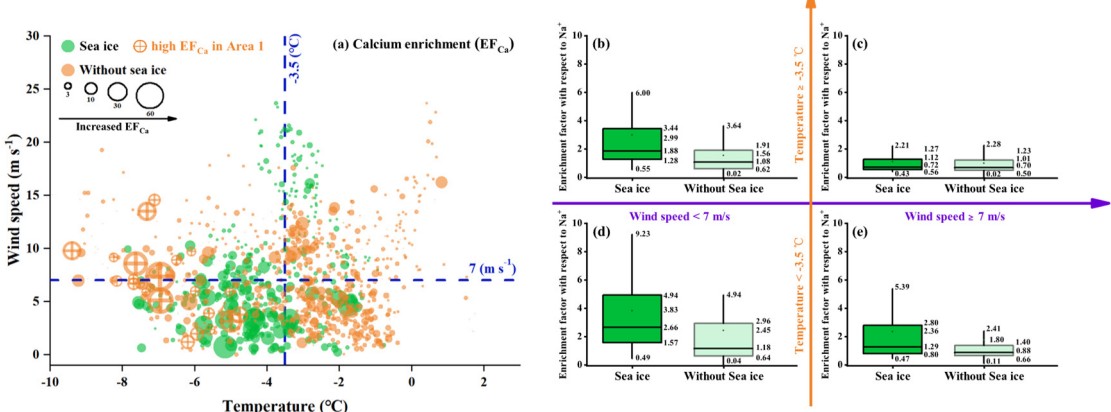

**Figure 3**
(a) Bubble chart of the hourly $Ca^{2+}$ enrichment factor ($EF_{Ca}$) with respect to $Na^+$ with different
environmental factors (ambient temperature, wind speed, and sea ice fraction). The green and
orange dots represent the $EF_{Ca}$ values for the periods with and without sea ice, respectively. The
orange marked dots represent a series of high $EF_{Ca}$ cases that were correlated with a high
concentration of chlorophyll-a during leg II of the cruise. (b)-(e) Data support of the bubble chart
represented by box and whisker plots. In the box and whisker plots, the marked values from top to
bottom are the 90th and 75th percentiles, mean, median, and 25th and 10th percentiles, respectively.

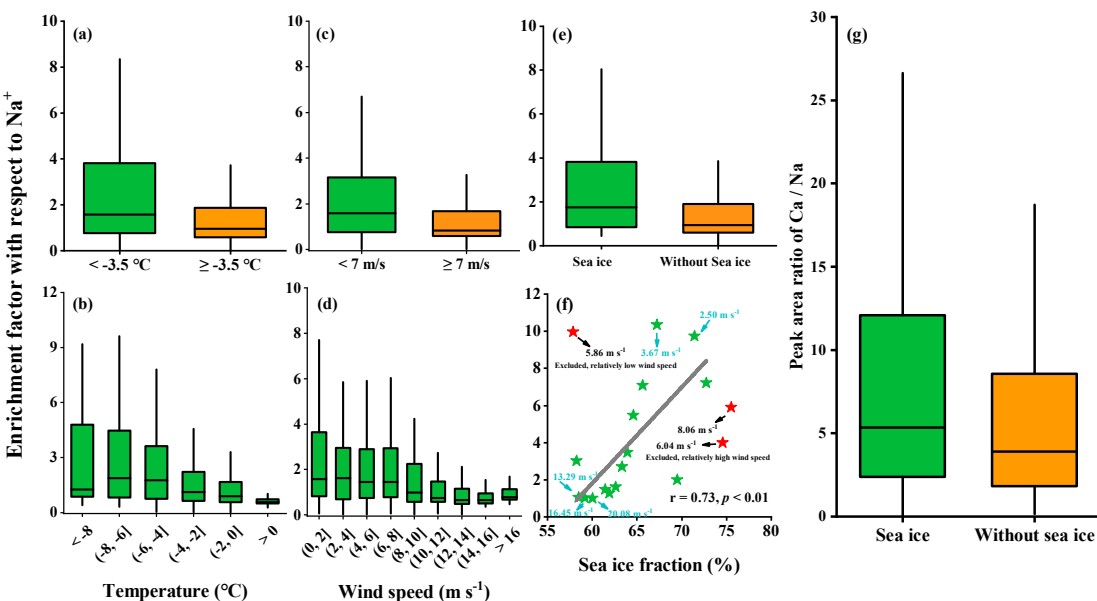


**Figure 4**
Enrichment factors of $Ca^{2+}$ with respect to $Na^+$ varied as a function of the ambient temperature (a-
b), wind speed (c-d), and sea ice fraction (e-f) during cruise observation campaigns. (g) A box and
whisker plot of the single-particle peak area ratio of Ca/Na in OC-Ca for the periods with and
without sea ice. In the box and whisker plots, the lower, median, and upper lines of the box denote
the $25^{th}$, $50^{th}$, and $75^{th}$ percentiles, respectively. The lower and upper edges denote the 10th and
90th percentiles, respectively. The black solid star (f) exhibited an anomalous trend due to its
nature of relatively high or low wind speed. The first point exhibited a high EF value because of
its relatively low wind speed (5.86 m s$^{-1}$). The second and third points exhibited low EF values
because of their relatively high wind speeds of 6.04 m s$^{-1}$ and 8.06 m s$^{-1}$, respectively. These three
points have been excluded from the correlation analysis.

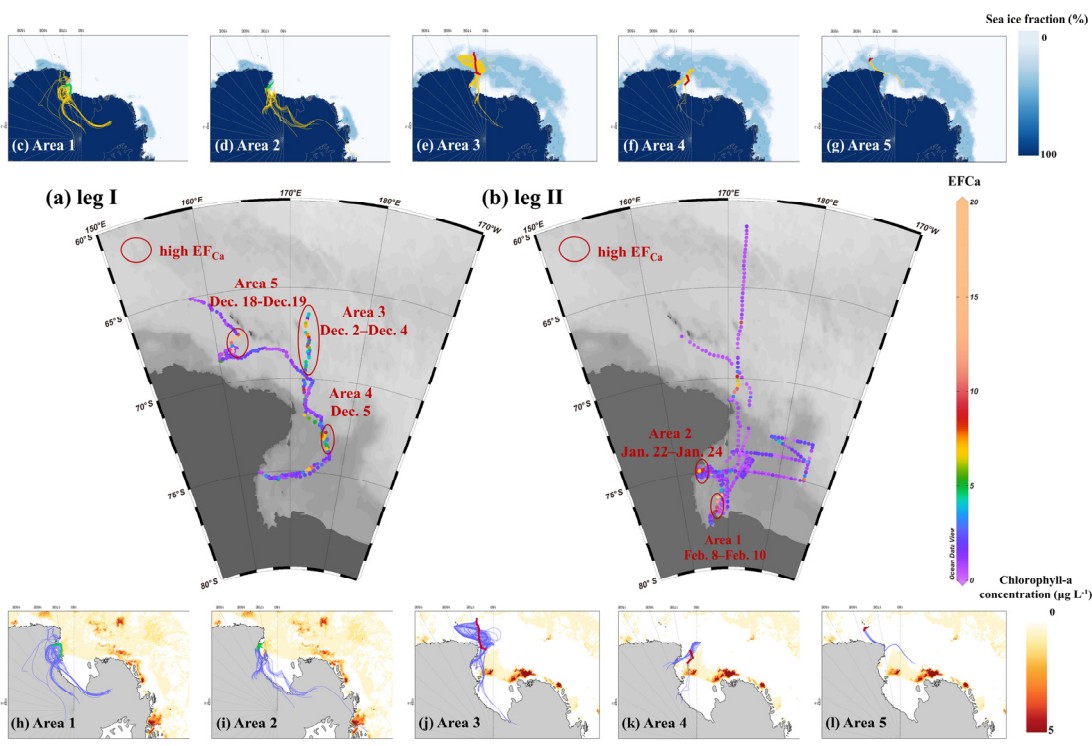

**Figure 5**
Distribution of $EF_{Ca}$ during (a) leg I and (b) leg II. Five distinct areas with continuous enhanced
$Ca^{2+}$ enrichment events, along with 96-hour back trajectories (one trajectory per hour in each
starting condition), sea ice fraction (c-g, yellow traces), and chlorophyll-a concentration (h-l, light-
blue traces). Lines in red and green referred to ship tracks for corresponding areas during leg I and
leg II, respectively.

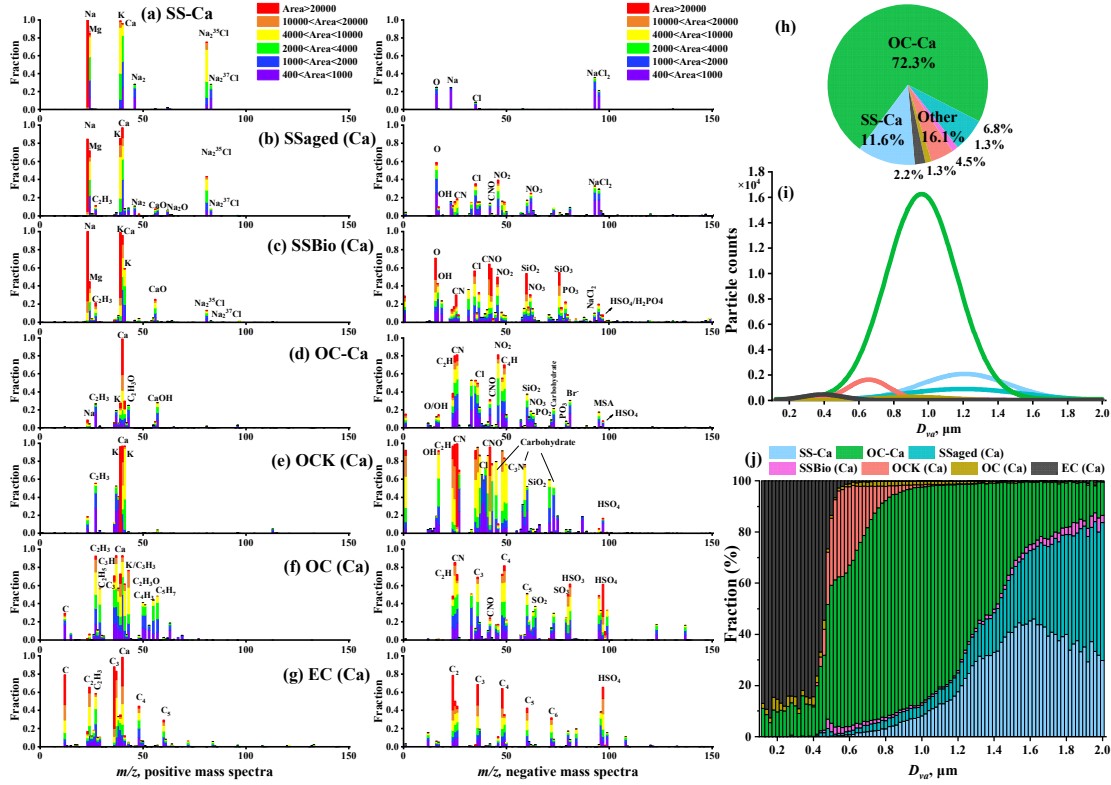


**Figure 6**

(a) – (g) Average digitized single-particle mass spectra of seven chemical classes of Ca-containing particles. New single-particle types are reclassified with *m/z* 40 [Ca²⁺] based on previous ART-2a results. (h) Relative proportion and (i) unscaled size-resolved number distributions of single-particle types using Gaussian Fitting. (j) Number fractions of single-particle types per size bin versus particle size.

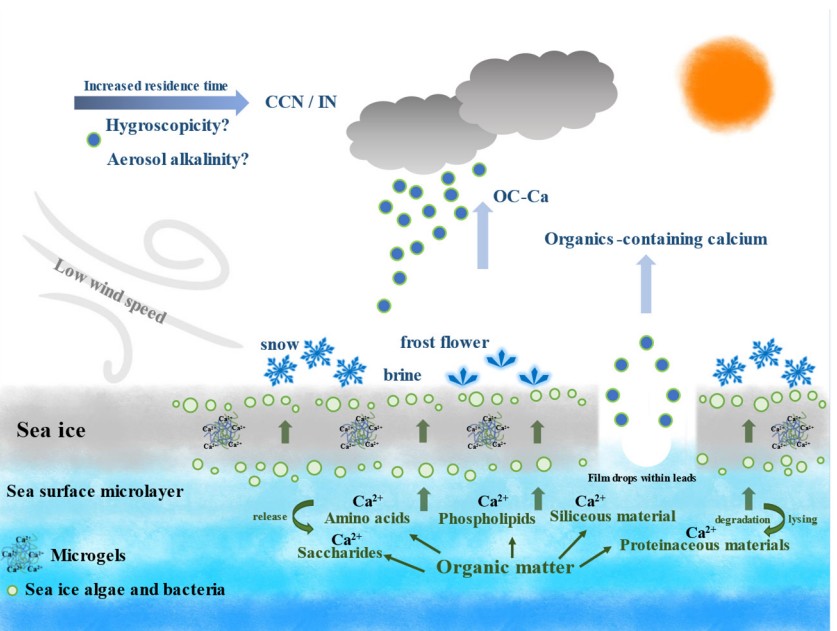

**Figure 7**

Schematic of the production of OC-Ca and its possible atmospheric implications beyond calcium

enrichment. $Ca^{2+}$ tends to bind with organic matter whining sea ice/seawater, and subsequently

assemble to marine microgels, likely present in the snow, frost flowers, and brine channels. With

the low wind-blown sea ice process and/or bubble bursting within sea ice leads, these gel-like

particles (i.e., OC-Ca) may be released to the Antarctic atmosphere, as a potential source of

CCN/IN. Notably, the dataset via SPAMS cannot directly identify marine microgels. OC-Ca was

likely associated with marine microgels, as calcium and biological organic material were

extensively internally mixed. This OC-Ca type has previously been observed in the laboratory

simulation of Collins et al. (2014).

**Table captions**

| EF$_{Ca}$ | Count (Correlation coefficient, r) | | | | Peak area (Correlation coefficient, r, $m/z$ 40 [Ca]$^{2+}$) | | | |
|---|---|---|---|---|---|---|---|---|
| | OC-Ca-Inorganic | OC-Ca-Organic | OC-Ca | SS-Ca | OC-Ca-Inorganic | OC-Ca-Organic | OC-Ca | SS-Ca |
| 0 - 5 | 0.08 | 0.31 | 0.18 | 0.07 | 0.18 | 0.44 | 0.41 | 0.04 |
| 5 - 10 | 0.15 | 0.37 | 0.27 | 0.04 | 0.14 | 0.36 | 0.33 | 0.06 |
| > 10 | 0.58 | 0.59 | 0.63 | 0.10[a] | 0.53 | 0.68 | 0.68 | 0.10 |
| Leg I | 0.45 | 0.59 | 0.55 | 0.02 | 0.53 | 0.60 | 0.60 | 0.03 |
| Leg II | 0.06 | 0.22 | 0.14 | 0.45 | 0.14 | 0.39 | 0.39 | 0.11 |
| Total | 0.28 | 0.51 | 0.42 | 0.21 | 0.31 | 0.51 | 0.49 | 0.03 |

a: $p$-value > 0.05
(Pearson method, two-tailed test)

**Table 1**

Correlation analysis between the OC-Ca (by count and by the peak area of $m/z$ 40 [Ca]$^+$) and its two subpopulations OC-Ca-Organic and OC-Ca-Inorganic, SS-Ca (by count and by the peak area of $m/z$ 40 [Ca]$^+$), and mass concentration of Ca$^{2+}$ in the variation of EF$_{Ca}$, with the $p$-value < 0.05.

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
