# Peer review of "Enrichment of calcium in sea spray aerosol: Insights from bulk measurements"

_EGUsphere, 2023_

## Author Comment (AC1)

Response to reviewer #1 comment on EGUSPHERE-2023-322 of Enrichment of calcium in sea spray aerosol through bulk measurements and individual particle analysis during the R/V Xuelong cruise over the Ross Sea, Antarctica

We would like to thank the reviewers for their valuable time, feedback, and comments. The suggested modifications have certainly improved the manuscript.

In this document, the review comments are shown in black. The author's response is shown in blue.

The revision is shown in red. Line numbers in the responses correspond to the revised manuscript with tracked change. All modifications can be found in the revised manuscript with tracked changes.

Su et al. conducted an analysis of the chemical composition of ambient aerosols collected during a research cruise in the Ross Sea, with a specific focus on calcium enrichment in sea spray aerosol.

However, their claims of providing insight into the calcium enrichment process in sea spray aerosols are not convincing for several reasons. Firstly, the authors cannot claim that they exclusively probed sea spray aerosol since their measurements of ambient aerosol contained other aerosols derived from various sources, including blowing snow and ice, long-range transported aerosols, and secondary aerosols formed from gaseous precursors. Therefore, it is recommended that this claim be removed from the title and text. Secondly, the authors' analysis is limited to correlating calcium enrichment with environmental variables such as wind speed and air temperature, without providing statistical or in-depth analysis of the meteorological or oceanographic conditions. Consequently, the manuscript reads more like a measurement report than a research article. Furthermore, to enhance the manuscript's readability, it is necessary to improve the writing structure at the sentence, paragraph, and section levels. Therefore, I believe that the work should be rejected in its current form and resubmitted as a measurement report. However, if the authors do wish to pursue publication of this work as an original research article, they should perform a more comprehensive analysis of their data and the conditions under which their measurements were made. Detailed suggestions for improvement are provided below.

Author's Response: We thank the anonymous reviewer for taking extensive time to carefully review our manuscript and providing valuable comments, which are of great advantage to the improvement of the manuscript.

For the first issue, we believe that the question of calcium enrichment in sea spray aerosol should only consider whether $Ca^{2+}$ and $Na^+$ are derived from sea spray aerosols (SSAs). We agree with the reviewer's comment that the bulk measurements contained various aerosol types, including those from wind-blown sea ice and/or snow upon sea ice. We suggest that they should be regarded as sea spray aerosols (Wagenbach et al., 1998; Rankin et al., 2000; Sander et al., 2006; Yang et al.,

2008; Rhodes et al., 2017; Yan et al., 2020a; Chen et al., 2022).

Our bulk measurement showed that the mass concentration of $Na^+$ was correlated well with that of $Cl^-$ (r = 0.99, $p < 0.001$) and $Mg^{2+}$ (r = 0.99, $p < 0.001$), indicating that they had the same source (i.e., sea spray). In contrast, the mass concentration of $Ca^{2+}$ was relatively weakly correlated with $Na^+$ (r = 0.51, $p < 0.001$), $Cl^-$ (r = 0.48, $p < 0.001$), and $Mg^{2+}$ (r = 0.53, $p < 0.001$). We believe that this difference is mainly due to the insolubility (or low water-solubility) of calcium salts (e.g., associated with carbonate ions and/or organic complexation). Of course, $Ca^{2+}$ may originate from dust of long-range transport and/or glacial dust. However, the back trajectory analysis and the mean mass concentration ratio of Ca/Na in the aerosol sample of 0.10 (lower than that in crust of 1.78, w/w) cannot effectively support the above inference. Additionally, we did not observe crustal mass spectral characteristics (e.g., -76 $[SiO_3]^-$, 27 $[Al]^+$, and 48 $[Ti]^+$/64 $[TiO]^+$) in the calcium-containing individual particles (Pratt et al., 2009; Zawadowicz et al., 2017). Regarding secondary aerosol formed from gaseous precursors, we suggest that it may be less relevant to calcium enrichment in SSAs. A possible particle type of secondary aerosols observed via SPAMS was OC(Ca), accounting for only a small proportion (1.3%). Taken together, we suggest that $Na^+$ and $Ca^{2+}$ measured in our bulk measurement shed light on the calcium enrichment of SSAs. We integrated the above discussion into the beginning of section 3.1. Please refer to the revised manuscript.

For the second issue, we agree with the reviewer's comment that the production of SSAs and subsequent species enrichment are associated with various meteorological or oceanographic conditions other than wind speed and/or temperature. Temperature, wind speed, and sea ice fraction were chosen, on the one hand, because the meteorological station onboard can provide accurate hourly resolution datasets of wind speed and temperature that matched ion concentrations. On the other hand, these environmental factors are associated with the yield of SSAs and calcium enrichment in SSAs (e.g., wind speed and temperature to the production of SSAs and sea ice fraction to calcium enrichment) (Hara et al., 2012; Forestieri et al., 2018; Zinke et al., 2022). Other meteorological or oceanographic conditions, such as seawater salinity, solar radiation, boundary layer height, total precipitation, etc., such as surface net solar radiation, snowfall, total cloud cover, surface pressure, total precipitation, boundary layer height, seawater salinity, etc., may also affect the calcium enrichment in SSAs through regulating the yield of sea salt (i.e., $Na^+$ mass concentration). However, they were not available in this study because of the lack of measurement during the cruise. In addition, the satellite data with low temporal-spatial resolution cannot match per hour in each starting condition. We hope that future research will further investigate the enrichment of specific species in SSAs under a wider range of meteorological or oceanographic conditions.

For the third issue, we appreciate the reviewer's valuable efforts in improving the manuscript's readability. In the revised manuscript, we first enhanced the readability of each sentence following the reviewer's comment. Then, we incorporated some of the important content and figures from the supporting information into the main text (e.g., the revised Figure 3). Lastly, we carefully checked the grammar, verb intense, and logical structure in the revised manuscript.

Lastly, we believe that our manuscript is suitable for publication as an original research article for two primary reasons: (1) We observed calcium enrichment in ambient aerosol samples in the Ross Sea, which could be associated with several environmental variables. While such an analysis based on bulk measurement may not be in-depth, few studies have established a relationship between the enrichment of specific species in aerosol samples and environmental factors. We hope our manuscript can serve as a modest spur to encourage future research to come forward with a
more thorough investigation of the enrichment of specific species in SSAs under different
meteorological or oceanographic conditions. (2) Insights into calcium enrichment of SSAs are not
exclusively derived from bulk measurement. Through the individual particle analysis, we observed
that a single-particle type of OC-Ca (internally mixed organics with calcium) accounted for the
largest proportion during the research cruise, and may be associated with calcium enrichment in
SSAs. We further hypothesize that the production mechanism of OC-Ca may be associated with
marine microgels based on its specific mixing state. In comparison with the mechanism of calcium
enrichment in aerosol samples, we suggest that the environmental behaviors of the possible gel-like
calcium particles (OC-Ca), such as their capacity to serve as could condensation nuclei (CCN)
and/or ice nuclei (IN) in the pristine Antarctic atmosphere, warrant more attention.
Detailed point-by-point responses are as follows:
Major points
1. The manuscript would benefit from improved writing clarity and readability. Specifically, the
authors could consider breaking down long sentences into shorter ones and using an active voice to
formulate their ideas more clearly. Verb tenses are also used inconsistently throughout the text, and
this should be addressed. In the minor points below, I provide specific examples to help the authors
improve their writing.
Author's Response: We would like to appreciate the reviewer for the valuable comment and
assistance in improving the readability and clarity of the manuscript. We have made the revisions
to enhance the readability of the article, according to the reviewer's detailed comments. In addition,
we carefully checked the verb tense and break down some long sentences throughout the manuscript
for better clarity. For detailed modification please refer to the minor comment's response and
revised manuscript.
2. In my opinion, the description of the methods used to obtain the samples is not adequately detailed.
Firstly, the text in section S2 should be integrated into the main manuscript. Secondly, I suggest
including statements about the efficiency of particle and gas collection in the sampler in the methods
section of the manuscript. While I assume this has been previously tested, it would be helpful to
have a statement such as "As shown in previous studies, 99% of particles entering the sampler were
retained on the impaction plate." If not all particle sizes are efficiently sampled, a statement about
the effective particle size ranges sampled by the system should be included instead. The same
applies to the gas phase sampling in the denuder. It would be helpful to know how effective it is and
what fraction of standard acid/base gases are sampled by this system.
Author's Response: Thanks for the reviewer's constructive suggestion. We have made the following
modifications based on your suggestions.
For the first point, we have integrated supplementary Text S2 into section 2.4.1 of the revised
manuscript to improve its coherence and accessibility.

For the second point, we would like to clarify that the trap efficiency of the $PM_{10}$ cyclone is greater than 99% for particles > 0.3 μm, $D_{a50} = 10 \pm 0.5$ μm. In addition, the collection efficiency of both aerosol and gas samples before entering the IC has been reported to be higher than 89% (for

0.056 μm particles, 89%; for 1 μm particles, 98%; for gas samples, > 90%) (Chang et al., 2007; Tian et al., 2017). The relevant descriptions have also been integrated into section 2.4.1 of the main text.

Please refer to the revised section 2.4.1 Aerosol water-soluble ion constituents.

3. I believe that the main manuscript would benefit from the inclusion of two key figures. Firstly, a map displaying the measurement stations would help readers orient themselves. Secondly, a schematic of the sampling system should also be included in the main manuscript. Additionally, it might be helpful to include a map showing the distribution of calcium enrichment as part of the results section to enhance the reader's understanding of the observations.

Author's Response: Thanks for the reviewer's constructive suggestion. We have incorporated three figures into the revised manuscript as suggested.

[Figure]

Fig. 1 Observation campaigns through R/V *Xuelong* in the Ross Sea, Antarctic. (a) Leg I took place from December 2-20, 2017. (b) Leg II was conducted from January 13 to February 14, 2018.

The first figure depicted the research cruise over the Ross Sea, Antarctic.

[Figure]

Figure 2 A schematic of the aerosol sampling system of IGAC and SPAMS during the research cruise over the Ross Sea, Antarctic.

The second figure is a schematic of the sampling system during the research cruise over the Ross Sea, Antarctic.

[Figure]

Figure 5 Distribution of $EF_{Ca}$ during (a) leg I and (b) leg II. Five distinct areas with continuous enhanced $Ca^{2+}$ enrichment events, along with 96-hour back trajectories (one trajectory per hour in each starting condition), sea ice fraction (c-g, yellow traces), and chlorophyll-a concentration (h-l, light-blue traces). Lines in red and green referred to ship tracks for corresponding areas during leg I and leg II, respectively.

The third figure included a map showing the distribution of $EF_{Ca}$ and events of enhanced calcium enrichment concerning 96-h back trajectories with sea ice fraction, and Chlorophyll-a concentration.

4. In my opinion, the content of text S1 appears to be more suitable for the introduction and discussion sections of the manuscript rather than as supplementary information. If the authors consider this information important for readers to understand their study, they should incorporate it into the main manuscript. The supplementary information should not be used for discussions that are integral to the main content of the study.

Author's Response: Thanks for the reviewer's constructive suggestions. Specifically, we have integrated the contents of supplementary Text S1 into the method, discussion, and conclusion sections of the revised manuscript.

Regarding the discussion of the IGAC dataset, we agree with your suggestion that it is more appropriate in section 2.4.1 Aerosol water-soluble ion constituents. We have made the necessary modifications to ensure a more cohesive and structured description.

Furthermore, we have incorporated discussions associated with the water-solubility of OC-Ca and its possible environmental behavior in the 4 Discussion and 5 Conclusion sections, respectively.

We believe that these changes have significantly improved the clarity and flow of the manuscript.

For detailed modifications please refer to Lines 440-445 in Section 4 Discussion and Lines540-

543 in Section 5 Conclusion.

5. Could the authors provide more details on how they prevented their research vessel from affecting their measurements? It is unclear whether they used a pollution control system or sector analysis to reduce potential ship emissions during their measurements. Additionally, while they used a PM10

cyclone to capture larger particles, it is not clear whether they took measures to eliminate the influence of smaller particles being emitted from the ship. To improve the clarity of their study, the authors should provide more information on these points.

Author's Response: Thanks for the reviewer's constructive suggestion. We apologize for the vague description of pollution control on the research vessel. Indeed, we have taken some measures to prevent any potential contamination during aerosol sampling. Firstly, the sampling inlet connecting to the monitoring instruments was fixed to a mast, located at the bow of the vessel, at a height of 20

m above the sea surface, which could prevent major contamination emitted from the chimney. In addition, the sampling inlet was fixed on a ship pillar with a rain cover, which could minimize the potential influence of violent shaking of the ship and sea waves. Secondly, we conducted the sampling only while the ship was sailing to avoid potential ship emissions under the low diffusion condition. Lastly, through individual particle analysis, we did not observe the mass spectral characteristics associated with ship emission (e.g., particles simultaneously contain $m/z$ 51 $[V]^+$, 67

$[VO]^+$, and element carbon) during the observation campaigns. We have clarified it in the revised manuscript as follows.

**2.3 Contamination control during observation campaigns**

During the research cruise, the major contamination source was identified as emissions from a
chimney located at the stern of the vessel and about 25 m above the sea surface. To mitigate the
potential impact of ship emissions on aerosol sampling, we have taken several measures. Firstly, a
total suspended particulate (TSP) sampling inlet connecting to the monitoring instruments was fixed
to a mast 20 m above the sea surface, located at the bow of the vessel. In addition, the sampling inlet
was fixed on a ship pillar with a rain cover, which could minimize the potential influence of violent
shaking of the ship and sea waves. Secondly, sampling was only conducted while the ship was
sailing, to avoid the possible effect of ship emission on aerosol sampling under the low diffusion
condition. Lastly, we did not observe the mass spectral characteristics associated with ship emission
(e.g., particles simultaneously contain $m/z$ 51 $[V]^+$, 67 $[VO]^+$, and element carbon) during the
observation campaigns (Liu et al., 2017; Passig et al., 2021). These measures ensured that the
collected data were representative and reliable for subsequent analysis.
6. The authors propose that blowing snow may be responsible for the observed Ca2+ enrichment in
the aerosols they collected. However, this contradicts their earlier assertion that their study provides
insights into Ca2+ enrichment in sea spray aerosol. Furthermore, their analysis to support this
conclusion lacks depth. While the association between sea-ice coverage and Ca2+ provides a clue,
the authors should conduct a more thorough investigation. For example, they could explore how
long a specific air mass they sampled spent over ice and examine correlations between wind speed
and sea-ice during sampled air-mass trajectories.
Author's Response: We appreciate the reviewer's valuable comment. As discussed above,
blowing-snow is also a potentially important source of sea spray aerosol (SSAs) (Yang et al., 2008).
And thus, the above statements are not contradictory. We agree with the reviewer's comment that
our analysis to support blowing-snow as a reason for calcium enrichment in SSAs is insufficient
due to the lack of datasets on the age of the snow (Yang et al., 2008) and the absence of
measurements of the chemical composition of the sea ice and snow. Thus, the discussion of the
potential impact of blowing-snow events on SSAs and calcium enrichment is limited. However, it
is important to note that snow lying on sea ice represents an important source not only of sea salt
(Yang et al., 2008) but also of bromide and marine organic aerosols (e.g., extracellular polymeric
substances (EPS), released from microorganisms) (Arrigo, 2014; Boetius et al., 2015). Coincidently,
the possible ion signals of bromide ($m/z$ -79 and -81) were also observed in OC-Ca, supporting a
potential source of blowing-snow. Hence, we suggest that blowing-snow may contribute to OC-Ca,
which in turn affects calcium enrichment in SSAs. In the original manuscript, such discussion is not
overextended due to the lack of additional strong evidence.
We conducted a further investigation into the relationship between enhanced calcium
enrichment events (Area 1-5) and 96-hour back trajectories. Our analysis revealed that these
enhanced calcium enrichment events were highly associated with air masses traveling over the ice
upon Ross Sea (marginal ice floe/sea ice) (>95%, by trajectory coverage) and/or Antarctic land
(land-based Antarctic ice, 57-59%). The back trajectory analysis also indicated that the open water and the long-range transport of dust were not responsible for the observed calcium enrichment in

SSAs. A detailed description has been integrated at the end of section 3.1.

As suggested by the reviewer, we examined the correlation between wind speed and sea ice fraction during leg I (sea ice period) and found a moderate negative correlation ($r = 0.50$, $p < 0.001$)

between them. Taken together, we believe that these analyses can further support our previous speculation that low wind-blown sea ice is responsible for calcium enrichment in SSAs. We have added the discussion into section 4, lines 463-467.

In this case, the calcium enrichment in SSAs could reasonably be attributed to the possible gel- like calcium-containing particles released by low-wind-blown sea ice. This inference is supported by the observation of air masses blown over a large fraction of sea ice/ land-based Antarctic ice, as well as a moderate negative correlation ($r = 0.50$, $p < 0.001$) between wind speed and sea ice fraction.

[Figure]

**Figure 5** Distribution of EF$_{Ca}$ during (a) leg I and (b) leg II. Five distinct areas with continuous enhanced Ca$^{2+}$ enrichment events, along with 96-hour back trajectories (one trajectory per hour in each starting condition), sea ice fraction (c-g, yellow traces), and chlorophyll-a concentration (h-l, light-blue traces). Lines in red and green referred to ship tracks for corresponding areas during leg

I and leg II, respectively.

| Item | Area 1 (2018.02.08 22:00-2017.02.10 22:00) | Area 2 (2018.01.22 17:00-2018.01.24 04:00) | Area 3 (2017.12.02 07:00-2017.12.04 19:00) | Area 4 (2017.12.05 00:00-2017.12.05 23:00) | Area 5 (2017.12.18 22:00-2017.12.19 05:00) | leg I | leg II | The whole observation |
|---|---|---|---|---|---|---|---|---|
| Duration (h) | 48 | 35 | 61 | 24 | 8 | 426 | 769 | 1195 |
| $EF_{Ca}$ | 10.13 ± 13.63 | 2.96 ± 2.12 | 5.47 ± 4.64 | 9.72 ± 18.75 | 30.98 ± 31.32 | 3.94 ± 8.50 | 2.11 ± 4.47 | 2.76 ± 6.27 |
| $EF_{K}$ | 2.88 ± 2.36 | 1.49 ± 0.77 | n.a. | 45.46 ± 14.79 | 1.22 ± 0.46 | 7.93 ± 14.03 | 1.67 ± 1.69 | 3.61 ± 8.45 |
| $EF_{Mg}$ | 2.88 ± 1.54 | 1.97 ± 0.69 | 7.89 ± 4.35 | 8.25 ± 2.90 | 1.38 ± 0.33 | 3.74 ± 3.75 | 1.80 ± 1.05 | 2.46 ± 2.53 |
| Temperature (°C) | -6.4 ± 1.2 | -2.9 ± 0.8 | -4.5 ± 0.9 | -4.0 ± 0.8 | -1.9 ± 2.2 | -4.1 ± 1.4 | -3.2 ± 2.2 | -3.5 ± 2.0 |
| Wind speed (m s$^{-1}$) | 5.7 ± 3.5 | 4.7 ± 1.8 | 6.04 ± 2.2 | 2.49 ± 1.1 | 5.1 ± 4.5 | 7.2 ± 5.5 | 7.1 ± 4.2 | 7.1 ± 4.7 |
| Sea ice fraction | 54.60 ± 0.02 | 54.53 ± 0.00 | 74.28 ± 1.41 | 71.41 | 58.06 ± 0.25 | 64.91 ± 5.57 | 54.59 ± 0.08 | 58.38 ± 6.07 |
| Chl-a concentration (µg L$^{-1}$) | 0.99 ± 1.65 | 0.10± 0.20 | Unavailable | Unavailable | Unavailable | 0.51 ± 0.29 | 0.44 ± 0.18 | 0.46 ± 0.23 |
| 96-Trajectory coverage (%)* Sea ice: Open water: Antarctic Land: | 28% 15% 57% | 33% 8% 59% | 95% 5% 2% | 95% 2% 3% | 96% 0% 4% | 92% 4% 4% | 30% 12% 58% | 52% 9% 39% |

Note: (1) Area 1 and 2 are divided during the leg II, whereas the Area 3, 4, and 5 are divided during the leg I. (2) The values of sea ice fraction and chl-a concentration present with daily resolution. Others present with hourly resolution. (3) No sea ice coverage is equivalent to the sea ice fraction below 55. (4) 96-Trajectory coverage (%)* correspounds to fraction of air masses traveled over different surface type when the peak EFCa value from Area 1 to 5.

**Table S4** Average enrichment factors for specific cations and metrological parameters over the different areas mentioned in Fig. 5 in main text. Although all areas exhibited significant $Ca^{2+}$ enrichment in SSAs, they may have varied due to synergetic environmental factors rather than a single factor. $Ca^{2+}$ enrichment in SSAs was notably observed with low wind speed, underscoring the effect of wind speed. The back trajectory coverage is labeled as sea ice, open water, and land. For leg I, the major positive $Ca^{2+}$ enrichment events were associated with Areas 3, 4, and 5. In addition to the lower wind speed, lower temperature, and the presence of sea ice, the air masses blowing over the large fraction of sea ice and marginal ice zone may play an important role in $Ca^{2+}$ enrichment. For the leg II, the major positive $Ca^{2+}$ enrichment events occurred in Areas 1 and 2, which mainly associated with lower wind speed and temperature. The air masses were mostly from the land-based Antarctic ice.

Minor points (1) Line 23 – I suggest this sentence is rephrased as follows: "Although calcium is known to be enriched in sea spray aerosols (SSA), the factors that control its enrichment remain ambiguous." Calcium can not have a mixing state – it is SSA that has a mixing state.

Author's Response: We agree with the reviewer's comment. It has been revised accordingly.

(2) Line 24 – I suggest the authors break this sentence in two to improve the clarity and readability and to make clear the research objectives and the source of data: "In this study, we examine how environmental factors affect the distribution of water-soluble calcium (Ca2+) in sea spray aerosols (SSAs). We obtained our data from observations taken during a research cruise on R/V Xuelong in the Ross Sea, Antarctica, from December 2017 to February 2018."

Author's Response: Many thanks for the reviewer's constructive suggestion. It has been revised accordingly.

(3) Line 27 – In order to improve the clarity of the sentence I suggest the authors use active voice and rephrase to make it a statement of the study's findings: "Our observations show that the enrichment of Ca2+ in aerosol samples is enhanced under specific conditions, including lower temperatures (< -3.5 ℃), lower wind speeds (< 7 m s-1), and the presence of sea ice."

Author's Response: Many thanks for the reviewer's constructive suggestion. It has been revised accordingly.

(4) I also suggest the use of "inaccurate" rather than "neglected" to more accurately describe the potential problem with current estimates of Ca2+ enrichment: "Our analysis of individual particle mass spectra revealed that a significant portion of calcium in SSA is likely bound with organic matter (in the form of a single-particle type, OC-Ca). This finding suggests that current estimates of Ca2+ enrichment based solely on water-soluble Ca2+ may be inaccurate."

Author's Response: Many thanks for the reviewer's constructive suggestion. It has been revised accordingly.

(5) Line 31 – I suggest the authors rephrase this sentence as a statement of the study's unique contribution: "Our study is the first to observe a single-particle type dominated by calcium in the Antarctic atmosphere."

Author's Response: Many thanks for the reviewer's constructive suggestion. It has been revised accordingly.

(6) Line 32 – I suggest the authors rephrase this sentence to clarify the specific aspect of the modeling that needs to be addressed and to make it a recommendation based on the study's findings: "Our findings suggest that future Antarctic atmospheric modeling should take into account the environmental behavior of individual OC-Ca."

Author's Response: Many thanks for the reviewer's constructive suggestion. It has been revised accordingly.

(7) Line 34 - I suggest the authors improve the clarity and readability of the sentence by rephrasing it as a statement of the study's importance: "With the ongoing global warming and retreat of sea ice, it is essential to understand the mechanisms of calcium enrichment and the mixing state of individual particles to better comprehend the interactions between aerosols, clouds, and climate during the Antarctic summer."

Author's Response: Many thanks for the reviewer's constructive suggestion. It has been revised accordingly.

In the revised manuscript, the abstract has been revised to:

**Abstract:** Although calcium is known to be enriched in sea spray aerosols (SSAs), the factors that affect its enrichment remain ambiguous. In this study, we examine how environmental factors affect the distribution of water-soluble calcium ($Ca^{2+}$) distribution in SSAs. We obtained our dataset from observations taken during a research cruise on the R/V *Xuelong* cruise in the Ross Sea, Antarctica, from December 2017 to February 2018. Our observations showed that the enrichment of $Ca^{2+}$ in aerosol samples was enhanced under specific conditions, including lower temperatures (< -3.5 ℃), lower wind speeds (< 7 m s$^{-1}$), and the presence of sea ice. Our analysis of individual particle mass spectra revealed that a significant portion of calcium in SSAs was likely bound with organic matter (in the form of a single-particle type, OC-Ca). Our findings suggest that current estimations of $Ca^{2+}$ enrichment based solely on water-soluble $Ca^{2+}$ may be inaccurate. Our study is the first to observe a single-particle type dominated by calcium in the Antarctic atmosphere. Our findings suggest that future Antarctic atmospheric modeling should take into account the environmental behavior of individual OC-Ca. With the ongoing global warming and retreat of sea ice, it is essential to understand the mechanisms of calcium enrichment and the mixing state of individual particles to better comprehend the interactions between aerosols, clouds, and climate during the Antarctica summer.

(8) Introduction in general - The text could benefit from using active voice to make it more engaging and easier to read. For example, instead of "The extent of enrichment and chemical signature of calcium may affect some physicochemical properties of SSA," use "Calcium enrichment and chemical signature can affect the physicochemical properties of SSA."

Author's Response: Many thanks for the reviewer's constructive suggestion. It has been revised accordingly.

(9) Introduction in general - Break up long sentences: Some sentences are quite long and complex, which can make them difficult to read and understand. Breaking them up into shorter, more concise sentences could help.

Author's Response: Many thanks for the reviewer's constructive suggestion. We carefully revised the introduction sentence by sentence. For example, Lines 77-80, "These results have greatly improved the understanding of the processes contributing to $Ca^{2+}$ enrichment, however, our understanding of how environmental factors synergistically affect such enrichment processes remains unclear" has been rephrased as "These results shed light on $Ca^{2+}$ enrichment process; however, our understanding of how environmental factors synergistically affect such enrichment process remains unclear." For more detailed modifications please refer to the revised introduction.

(10) Introduction in general – Use consistent verb tense and try to present conclusions first and then the sources that support the conclusions to make it easier for readers to follow. For example Line 58, "A growing number of studies have shown that calcium (Ca2+) is significantly enriched in SSA relative to bulk seawater (Table S1) (Keene et al., 2007; Hara et al., 2012; Cochran et al., 2016; Salter et al., 2016; Cravigan et al., 2020; Mukherjee et al., 2020)" could be rephrased as "Several studies have demonstrated a significant enrichment of calcium (Ca2+) in SSA compared to bulk seawater, as presented in Table S1 and documented by Keene et al. (2007), Hara et al. (2012), Cochran et al. (2016), Salter et al. (2016), Cravigan et al. (2020), and Mukherjee et al. (2020)."

Author's Response: Many thanks for the reviewer's constructive suggestion. It has been revised
accordingly.

(11) Introduction in general - Define acronyms when they are first used: Some acronyms are used
without being defined, which can be confusing for readers who are not familiar with the field. For
example, "SSA" is used multiple times without being defined until later in the text.

Author's Response: Many thanks for the reviewer's constructive suggestion. It has been revised
correspondingly.

(12) Introduction in general - Some points could be clarified or expanded upon to help readers
understand the context better. For example, on line 53 what do the authors mean by "the most
efficient gelling agent"?

Author's Response: We apologize for this ambiguity. We would like to express that $Ca^{2+}$ can readily
induce the gelation of organic matter as the most efficient gelling agent (Carter-Fenk et al., 2021).
To avoid the unclarity, this sentence has been revised, please refer to Lines 61-64 in the manuscript.
Calcium is one of the components of SSA, which can present as inorganic calcium (e.g., $CaCl_2$ and
$CaSO_4$) (Chi et al., 2015) as well as organic calcium (i.e., $Ca^{2+}$ can readily induce the gelation of
organic matter, presenting as the most efficient gelling agent) (Carter-Fenk et al., 2021).

(13) Line 69 – This sentence does not read well and it is unclear exactly what the authors mean. I
assume that what the authors mean is that our current understanding of the enrichment of Ca2+ in
SSA is the result of measurements of only water-soluble Ca2+. If that is the case the authors need
to make this point plainly. The authors then need to inform the reader of what alternatives there are.
Presumably, measurement approaches that determine not only the amount of water-soluble Ca2+
but also insoluble Ca2+ in the form of calcareous shell debris or the like could be used. Here would
be a good point to outline the difference clearly.

Author's Response: We appreciate the reviewer' comment.

Here we would like to present the current understanding of calcium enrichment in SSAs based
on its chemical form. By comparison of the two hypotheses proposed, we would like to emphasize
the significance of the chemical form of calcium to atmospheric chemistry. Then, we propose that
current estimations of $Ca^{2+}$ enrichment may be inadequate due to the inconsideration of insoluble
$CaCO_3$ and low water-soluble organically complexed calcium. Lastly, we suggest that an alternative,
such as individual particle analysis, may provide new insight into calcium enrichment and its
chemical form. We have revised this section in response to the reviewer's comment.

To date, a unified consensus on the chemical form of calcium to explain calcium enrichment in
SSAs has not been reached. Two hypotheses have been proposed: (i) Calcium enrichment is
dominated by inorganic calcium, such as $CaCO_3$ and $CaCl_2$. $Ca^{2+}$ is enriched close to the seawater
surface in the form of ionic clusters (most probably with carbonate ions) (Salter et al., 2016).
Another source of $CaCO_3$ is directly from calcareous shell debris (Keene et al., 2007). Through bubble bursts, both $CaCO_3$ and $CaCl_2$ along with sea salt can be emitted into the atmosphere. In addition, the sea salt fractionation by precipitation of ikaite ($CaCO_3 \cdot 6H_2O$) may contribute to calcium enrichment in aerosol during the freezing of sea ice (Hara et al., 2012).(ii) Calcium enrichment is attributed to organically complexed calcium. $Ca^{2+}$ may bind with organic matter, which is relevant with marine microgels and/or coccolithophore phytoplankton scales, and can be emitted by bubble bursting (Oppo et al., 1999; Sievering, 2004; Leck and Svensson, 2015; Cochran et al., 2016; Kirpes et al., 2019; Mukherjee et al., 2020). The chemical form of calcium can determine its atmospheric role. Inorganic calcium may exhibit stronger aerosol alkalinity and hygroscopicity than organic calcium (Salter et al., 2016; Mukherjee et al., 2020). However, current estimations of calcium enrichment based solely on water-soluble $Ca^{2+}$ may not precisely explain the calcium distribution in SSAs. This is because the amount of low water-soluble complexation of $Ca^{2+}$ with organic matter (e.g., aged $Ca^{2+}$-assembled gel-like particles) (Orellana and Verdugo, 2003; Leck and Bigg, 2010; Russell et al., 2010; Orellana et al., 2011; Leck and Svensson, 2015) and insoluble $Ca^{2+}$ in the form of calcareous shell debris or the like may not be considered. Thus, an alternative method, such as discerning the mixing state based on single-particle analysis, may provide unique insights into the chemical form of calcium, and thus the mechanisms of calcium enrichment in SSAs.

(14) Line 70 – I would argue that the authors need to do a better job of describing the two hypotheses for why Ca2+ may be enriched in SSA. As they state one possible mechanism is the complexing of Ca2+ to organic matter. A second possible mechanism Ca2+ ions are enriched close to the water surface in the form of ionic clusters most probably with carbonate ions.

Author's Response: Thanks for the reviewer's comments. We have further improved the related description of the two hypotheses of calcium enrichment according to your suggestion. Please refer to the reply of (13).

(15) Methods section 2.1 – I suggest the authors separate the information in this section into three distinct paragraphs to improve organization. E.g. one starting "Our study focused on the Ross Sea region of Antarctica (50 to 78° S, 160 to 185° E) (see Fig. S1), where we conducted two separate observation campaigns aboard the R/V Xuelong…". A second paragraph starting "The first observation campaign (Leg I) took place from December 2-20, 2017, during the sea ice period. The second campaign (Leg II) was conducted…" and a third starting "The sampling design for Leg I and Leg II aimed to investigate the differences in atmospheric aerosol characteristics…" etc.

Author's Response: Many thanks for the reviewer's constructive suggestion. It has been revised accordingly.

**2.1 The R/V *Xuelong* cruise and observation regions**

Our study focused on the Ross Sea region of Antarctica (50 to 78° S, 160 to 185° E) (**Fig. 1**), where we conducted two separate observation campaigns aboard the R/V *Xuelong*. During the observations, this region was relatively isolated from the impact of long-range transport of anthropogenic aerosols and has experienced the sea ice retreat (Yan et al., 2020a).

The first observation campaign (Leg I) took place from December 2-20, 2017, during the sea ice period. The second campaign (Leg II) was conducted from January 13 to February 14, 2018, during the period without sea ice. The sampling design for Leg I and Leg II aimed to investigate how changing environmental factors affect the enrichment extent of calcium and the characteristics of individual particles.

(16) Methods section 2.1 – I suggest the authors use more precise terminology (e.g., "observation campaigns" instead of "observations carried out"; "sampling design" instead of "sampling")

Author's Response: It has been revised as suggested.

(17) Methods section 2.1 – I suggest the authors remove redundant or unnecessary phrases, such as

"hereafter" and "when the ocean was covered by sea ice" (since this is already clear from the "sea ice period" description).

Author's Response: It has been revised as suggested.

(18) Methods section 2.2 – Again I think it would help clarity if the authors rephrased using active voice e.g. "We measured various meteorological parameters, such as ambient temperature, relative humidity (RH), wind speed, and true wind direction using an automated meteorological station located on the top deck of the R/V Xuelong…" and "To determine the type of air masses, we used the NOAA Hybrid Single-Particle Lagrangian Integrated Trajectories (HYSPLIT, version 4.9)

model to perform 72-hour back trajectory analysis…" and "Additionally, we obtained the monthly sea ice fraction from the Sea Ice Concentration Climate Data Record with a spatial resolution of 25

km…" etc.

Author's Response: Many thanks for the reviewer's constructive suggestion. These sentences have been revised accordingly.

**2.2 Meteorological parameters and satellite data of air masses, sea ice, and chlorophyll-a**

We measured various meteorological parameters, such as ambient temperature, relative humidity (RH), wind speed, and true wind direction using an automated meteorological station located on the top deck of the R/V *Xuelong* (**Fig. S1 and Table S2**).

To determine the type of air masses, we first overviewed the 72-hour back trajectory with daily resolution per each starting location by using the NOAA Hybrid Single-Particle Lagrangian

Integrated Trajectories (HYSPLIT, version 4.9) model (**Fig. S2**). Additionally, we conducted a 96- hour back trajectory analysis with an hourly resolution, which covered the enhanced calcium enrichment events associated with sea ice fraction and chlorophyll-a concentration (discussed in section 3.1), using the TrajStat in Meteoinfo (version 3.5.8) (Wang et al., 2009; Wang, 2014).

Meteorological data used for back trajectory analysis obtained from the Global Data Assimilation

System (GDAS, ftp://ftp.arl.noaa.gov/pub/archives). Moreover, we obtained the monthly sea ice fraction from the Sea Ice Concentration Climate Data Record with a spatial resolution of 25 km (https://www.ncei.noaa.gov/products/climate-data-records/sea-ice-concentration) and the 8-day chlorophyll-a concentration from MODIS-aqua with a spatial resolution of 4 km (https://modis.gsfc.nasa.gov) (**Fig. S3**).

During the R/V *Xuelong* cruise observation campaigns, leg I was dominantly affected by the air masses from the sea ice-covered open water (92%), and leg II was mainly affected by the air masses from continental Antarctica (58%) (**Table S2**). The average ambient temperature (-4.0 ± 1.4 °C

vs. -3.1 ± 2.2 °C), wind speed (7.2 ± 5.5 m s$^{-1}$ vs. 7.1 ± 4.2 m s$^{-1}$), and chlorophyll-a concentration (0.51 ± 0.29 μg L$^{-1}$ vs. 0.44 ± 0.18 μg L$^{-1}$) varied slightly between legs I and II (**Table S2**).

(19) Line 137 - Always be specific. As such this would read better as: "Throughout the observations, the mean Na+ and Ca2+ mass concentrations were 364.64 ng m-3 (ranging from 6.66 to 4580.10 ng m-3) and 21.20 ng m-3 (ranging from 0.27 to 334.40 ng m-3), respectively, which were more than

10 times higher than the detection limits."

Author's Response: Many thanks for the reviewer's suggestion. It has been revised as suggested.

(20) Line 140 – The sentence makes no sense to me. How can wind speed alone be used to rule out the potential influence of the research vessel on their measurements?

Author's Response: We apologize for the ambiguity in our previous statement. We would like to express that the low wind speed may not facilitate the dispersion of ship emissions. We have rephrased it into the new section 2.3 of contamination control during observation campaigns.

(21) Line 180 – You need to state that 0.038 is the molar ratio and not the mass ratio of Ca2+ to Na+

since the latter is approximately 0.066 (for every gram of sodium in seawater, there are about 0.066

grams of calcium).

Author's Response: Thanks for the reviewer's comment. We have carefully checked the mass and molar ratio of Ca$^{2+}$ to Na$^+$ in seawater, respectively. The mass ratio of Ca$^{2+}$ to Na$^+$ in seawater is

~0.038 (400[Ca]$^{2+}$/10561[Na]$^+$), while the molar ratio of Ca$^{2+}$ to Na$^+$ in seawater is ~0.022

(0.01[Ca]$^{2+}$/0.459174[Na]$^+$) (Bowen, 1979; Boreddy and Kawamura, 2015; Azadiaghdam et al.,

2019). We have also provided a link to a web resource: (URL:

https://web.stanford.edu/group/Urchin/mineral.html) for further information. We have clarified it.

Please refer to Line 285.

---

## Author Comment (AC2)

Response to reviewer #2 comment on EGUSPHERE-2023-322 of Enrichment of calcium in sea spray aerosol through bulk measurements and individual particle analysis during the R/V Xuelong cruise over the Ross Sea, Antarctica

We would like to thank the reviewers for their valuable time, feedback, and comments. The suggested modifications have certainly improved the manuscript.

In this document, the review comments are shown in black. The author's response is shown in blue.

The revision is shown in red. Line numbers in the responses correspond to the revised manuscript with tracked change. All modifications can be found in the revised manuscript with tracked changes.

This manuscripts present data on Ca enrichment in sea spray aerosol. This is an interesting topic in atmospheric research because of the CCN activation of sea spray aerosol particles. The mechanism of Ca enrichment is not yet fully understood. The data set itself (from a ship cruise at Antarctica) is rare and valuable.

However, there are some weaknesses in the presentation and interpretation of the results. Especially the single particle results are presented as measured, but connection to the main question (i.e. the mechanism behind the Ca enrichment) is not made clear. I don't think that it is unexpected that Ca and organics are internally mixed in sea spray particles. However, how the organic Ca helps to explain the Ca enrichment is not clear to me.

Thus, I think that there are major revisions necessary before the manuscript can be accepted for ACP.

My comments and concerns are listed in the following:

Author's Response: We would like to express our gratitude to the anonymous reviewer for the thorough review of our manuscript and for providing insightful comments that have significantly contributed to its quality.

To better understand calcium enrichment in sea spray aerosols (SSAs), it is necessary to verify the chemical form of calcium. This is because the current water-soluble estimation of $Ca^{2+}$

enrichment in SSAs may be inaccurate without considering organically complexed calcium. Thus, we believe that single-particle analysis is a crucial aspect of comprehending the mechanism of calcium enrichment in SSAs. In this study, we identified a single-particle type that calcium internally mixed with organics (i.e., OC-Ca). It should be noted that OC-Ca is not equal to organic

Ca. Although we cannot directly identify the chemical form of calcium via SPAMS, we rigorously inferred that OC-Ca may be organically complexed calcium based on its specific mixing state and thus be partially water-soluble.

We attempted to explain the mechanisms of calcium enrichment in SSAs by establishing a relationship between IGAC and SPAMS datasets. Initially, we found calcium enrichment in ambient aerosol samples in the Ross Sea, which we hypothesize is associated with several environmental variables, such as sea ice fraction, ambient temperature, and wind speed. Then, we attempted to investigate which particle types of calcareous aerosol contribute to calcium enrichment by comparing the chemical composition (e.g., the peak intensity of Ca), mixing state, and size of individual aerosols using SPAMS, as well as the absolute mass concentration using IGAC. Our results suggested that a single-particle type of OC-Ca (internally mixed organics with calcium) may partially contribute to calcium enrichment in SSAs.

Based on a theory of strong coordination between $Ca^{2+}$ and organic matter and the specific mixing state of OC-Ca observed in the Ross Sea, we further infer that the production mechanism of OC-Ca may be associated with marine microgels. Therefore, a comprehensive understanding of the characteristics of OC-Ca behind the mechanisms of calcium enrichment is conducive to further recognizing the CCN and IN activation in remote marine areas.

Detailed point-by-point responses are as follows:

General comments

1. Manuscript structure:

The main text of the manuscript is rather short. Many of the important and interesting discussions regarding methods and uncertainties are only found in the supplement. This is unsual for an ACP manuscript and make me suspect that the manuscript had been originally submitted somewhere else.

Author's Response: Thanks for the reviewer's comment. We would like to clarify that this manuscript has undergone extensive revision before submission to Atmospheric Chemistry and Physics. And we are sorry for the unclear. To improve writing structure and readability, we have incorporated the following parts into the main text, as suggested by reviewer 1# and reviewer 2#.

By incorporating these additions, we believe that the revised manuscript has achieved a more comprehensive and cohesive structure, and we thank the reviewer for their helpful input in this regard.

I recommend moving the following parts of the supplement into the main text:

S1: merge with section 4

Author's Response: Thanks for the reviewer's constructive suggestion. We have incorporated Text S1 into Section 2.4.1 Aerosol water-soluble ion constituents, Section 4 Discussion, and Section 5 Conclusions and atmospheric implications.

S2: merge with section 2.3

Author's Response: We have incorporated Text S2 into Section 2.4.

from S3: lines 123 to 141: move to section 4

Author's Response: We have revised the manuscript as suggested. Please refer to the ending of section 4 Discussion (Lines 475-510).

S6: move to section 3.2

Author's Response: Thanks for the reviewer's constructive suggestion. Some of the content in Text

S6 (lines 211-236) has been present in the original main text (lines 267-305 in Section 4). Therefore, we incorporated lines 203-210 into the beginning of Section 3.1.

We propose that both $Na^+$ and $Ca^{2+}$ in our observations originated from marine sources. The mass concentration of $Na^+$ exhibited a strong positive correlation with that of $Cl^-$ ($r = 0.99$, $p < 0.001$)

and $Mg^{2+}$ ($r = 0.99$, $p < 0.001$) (**Fig. S6**), indicating that they had a common origin (i.e., sea spray).

However, it is not surprising that the mass concentration of $Na^+$ showed a relatively weak correlation with that of $Ca^{2+}$ ($r = 0.51$, $p < 0.001$) (**Fig. S6**). This can be explained by the low water-soluble complexation of $Ca^{2+}$ with organic matter and/or insoluble $Ca^{2+}$ in the form of calcareous shell debris, such as $CaCO_3$. In addition, the potential impact of long-range transport of anthropogenic aerosols and dust contributing to $Ca^{2+}$ may be limited due to the predominance of polar air masses during the observation campaigns (see **Fig. S1**).

S8: move to main text, maybe as an additionally results subsection

Author's Response: Thanks for the reviewer's suggestion. We have carefully considered the results that there is little difference between leg I and leg II regarding the chemical composition, size, and mixing state of OC-Ca particles. Therefore, we hope to keep this part in the supporting information.

Also, we give a summary of this part in the end Section 2.4.2 Single-particle analysis.

There was little difference in individual particle analysis regarding chemical composition, size, and mixing state of particle clusters obtained from leg I and leg II (SI Text S3).

Taken together, we believe that these additions have significantly enhanced the clarity and organization of our manuscript and would like to express our appreciation to the reviewer for their helpful suggestion. Please refer to the revised manuscript for more details.

2. Calcium enrichment:

It is known that calcium is enriched in sea spray. The data from the IGAC instrument confirm this nicely. The results show that the highest enrichment factors are found at low temperate, low wind speed and sea ice conditions. This is a solid result, but I am no expert in this field and can not judge whether this is new or not.

Author's Response: Thanks for the reviewer's comments.

Calcium enrichment in SSAs has been compellingly verified in previous studies. We provided a summary of recent advances in calcium enrichment in SSAs in Table S1. These studies indeed demonstrated the presence of calcium enrichment in SSAs, but they have not established a clear relationship between calcium enrichment and various environmental factors.

In this study, we discussed how specific environmental factors affect calcium enrichment through field observations. We believe that these interesting findings could provide a better understanding of the mechanisms behind calcium enrichment in SSAs.

3. Single-particle analysis:

The abstract suggests that the controlling factors of the Ca enhancement wich are still unknown are studied in this manuscript.

It is not clear to me what the results of the mansucript mean for the controlling factors.

Author's Response: We apologize for this misleading. Here "control" may not be appropriate. We would like to express that calcium enrichment in SSAs could be affected by a series of environmental factors. Our results indicated that the enhanced $Ca^{2+}$ enrichment in SSAs was sensitive to the lower temperature, lower wind speeds, and the presence of sea ice. To avoid the potential ambiguities, we have rephrased the abstract as follows:

**Abstract:** Although calcium is known to be enriched in sea spray aerosols (SSAs), the factors that affect its enrichment remain ambiguous. In this study, we examine how environmental factors affect the distribution of water-soluble calcium ($Ca^{2+}$) distribution in SSAs. We obtained our dataset from observations taken during a research cruise on the R/V *Xuelong* cruise in the Ross Sea, Antarctica, from December 2017 to February 2018. Our observations showed that the enrichment of $Ca^{2+}$ in aerosol samples was enhanced under specific conditions, including lower temperatures (< -3.5 ℃), lower wind speeds (< 7 m s$^{-1}$), and the presence of sea ice. Our analysis of individual particle mass spectra revealed that a significant portion of calcium in SSAs was likely bound with organic matter (in the form of a single-particle type, OC-Ca). Our findings suggest that current estimations of $Ca^{2+}$

enrichment based solely on water-soluble $Ca^{2+}$ may be inaccurate. Our study is the first to observe a single-particle type dominated by calcium in the Antarctic atmosphere. Our findings suggest that future Antarctic atmospheric modeling should take into account the environmental behavior of individual OC-Ca. With the ongoing global warming and retreat of sea ice, it is essential to understand the mechanisms of calcium enrichment and the mixing state of individual particles to better comprehend the interactions between aerosols, clouds, and climate during the Antarctica summer.

A calcium-dominated OC-Ca particle type is detected. This particle type dominates the Ca- containing particle types, but it is not clear if these particles really represent microgels. The process how the biological organic material and the calcium end up in the same particle can not be identified from SPMS data alone.

Author's Response: We agree with the reviewer's comments regarding the limitations of the SPMS

dataset to demonstrate that the particle type of OC-Ca represents microgels.

Based on the specific mixing state of OC-Ca, we infer that the OC-Ca might be marine microgels. As previously reported, on the one hand, $Ca^{2+}$ tends to bind with organic matter of biogenic origin, such as exopolymer substances (EPSs), and subsequently assemble as marine microgels (Verdugo et al., 2004; Gaston et al., 2011; Krembs et al., 2011; Orellana et al., 2011; Verdugo, 2012; Orellana et al., 2021). On the other hand, Leck, Bigg, and their colleagues have reported the presence of microgels in the cloud samples in the polar region (Leck and Bigg, 2005a, b; Bigg and Leck, 2008; Leck and Bigg, 2010; Leck et al., 2013; Kirpes et al., 2019). We cannot accurately identify whether the OC-Ca is microgels. We have clarified it in section 4 to better highlight this limitation. Please refer to lines 630-633.

Notably, the dataset via SPAMS cannot directly identify marine microgels. OC-Ca was likely associated with marine microgels, as calcium and biological organic material were extensively internally mixed. This OC-Ca type has previously been observed in the laboratory simulation of Collins et al. (2014).

4. Particle size range:

The difference in the size rane of the two techniques is only discussed in the supplement. This is an important point when it comes to comparing the results of the two techniques. I suggest to move this part (supplement lines 123 - 141) into the main text (see comments above).

Author's Response: Thanks for the reviewer's constructive suggestion. We have incorporated lines 123 to 141 of Text S3 into the ending of Section 4 Discussion. Please refer to lines 492-501 in the revised manuscript.

Furthermore, it is not only the size range, but also a comparison between number fractions or peak intensities (SPMS) and mass concentrations (IGAC). This should also be discussed.

Author's Response: Many thanks for the reviewer's constructive suggestion. We conducted a comparison analysis between particle count, peak area, and mass concentration, as shown in Table 1. The discussion of this part was also incorporated into Section 4, such as lines 402-404, 414-422, 423-427, etc. Meanwhile, we have moved Table S5 to the main text as Table 1.

The OC-CA particle size distribution as displayed (Fig 3 i) centers around 1 μm, but this may be an instrumental effect (transmission and detection efficeny of the SPAMS). A plot of particle number fractions per size bin versus particle size will better show the contribution of each particle type.

Author's Response: Thanks for the reviewer's constructive suggestion. As mentioned by the reviewer, the particle size distribution may be affected by the transmission efficiency of the SPMS. We have added a subplot of particle number fractions per size bin versus particle size into Fig. 3 (j), as follows:

[Figure]

Figure 6

(a) – (g) Average digitized single-particle mass spectra of seven chemical classes of Ca-containing particles. New single-particle types are reclassified with *m/z* 40 [Ca$^{2+}$] based on previous ART-2a results. (h) Relative proportion and (i) unscaled size-resolved number distributions of single-particle types using Gaussian Fitting. (j) Number fractions of single-particle types per size bin versus particle size.

5. Trajectories:

More trajectories are needed. Either finer time steps (one trajectory per hour) or the "ensemble mode" of HYSPLIT, where many trjectories are calculated per start time with slight differences in the starting conditions. Only through such ensembles one can see the variations between individual trajectories and can judge the reliability of the backward calculation.

Author's Response: Many thanks for the reviewer's constructive suggestion. As suggested, we conducted a 96-hour back trajectory analysis, which includes one trajectory per hour in each starting condition. This analysis covered the selected enhanced calcium enrichment events (Area 1-5) and incorporated surface types such as sea ice, open water, Antarctic land, and chlorophyll-a concentration. Our results indicated that air masses traveling over the ice upon Ross Sea (marginal ice floe/sea ice) and/or Antarctic land were highly associated with enhanced calcium enrichment events. Moreover, the analysis indicated that the long-range transport of dust and open water are unlikely responsible for the observed calcium enrichment in SSAs. The discussion of this analysis has been incorporated into section 3.1 of the main text. We have also supplemented a new plot,

Figure 5, to illustrate the results of the 96-hour back trajectory analysis.

[Figure]

Figure 5

Distribution of EFCa during the (a) leg I and (b) leg II. Five distinct areas with continuous enhanced

Ca$^{2+}$ enrichment events, along with 96-hour back trajectories (one trajectory per hour in each starting condition), sea ice fraction (c-g, yellow traces), and chlorophyll-a concentration (h-l, light- blue traces). Lines in red and green referred to ship tracks for corresponding areas during the leg I

and leg II, respectively.

Specific points

Section 3.2

(1) What do you mean by "were further refined with an ion signal of m/z 40 [Ca]+."? Was there any threshold applied to the intensity of m/z 40 or were all mass spectra selected that had a signal > 0 at m/z 40? Please explain. >0

Maybe this should be explained already in section 2.3.2 (lines 160-170)?

Author's Response: We apologize for any confusion. In the data analysis of SPAMS, we first applied the ART-2a algorithm to cluster the obtained particles into seven groups. We then reclassified the clustered groups by setting a threshold for $m/z$ 40 [Ca]$^+$ signal (> 0) to obtain individual calcareous particles. This means that all particles that were reclassified contained signals of $m/z$ 40 [Ca]$^+$. We hope this clears up any misunderstandings and have revised the relevant

 sentences as follows.

 To elucidate the mixing state of individual calcareous particles, we set a threshold of *m/z* 40 [Ca]$^+$

 to reclassify all single-particle types that were obtained from the ART-2a algorithm. This means that

 all reclassified particles contain signals of *m/z* 40 [Ca]$^+$.

(2) Figure 3:

i) the size distributions look like Gauss fits. Please confirm if they are. I suggest using log scales for both axes. For the diameter axis, this is very common in aerosol science, and for the y-axis it will make particle types with low concentrations better visible. On a log scale, the Gauss fits should be replaced by lognormal fits.

Author's Response: Thanks for the reviewer's constructive suggestion. We agree with the reviewer's comment that using a double logarithmic axis is common in aerosol science. However, redrawing the plot using this axis format may not be suitable or visually effective. Therefore, we have followed the above suggestion and created a plot of particle number fractions per size bin versus particle size. We used Gaussian fitting to analyze the size distribution, which we have confirmed in the figure caption.

[Figure]

A plot of size distributions using a double logarithmic axis.

Minor points (3) Title: remove the "through", e.g.:

Enrichment of calcium in sea spray aerosol: Bulk measurements and individual particle analysis during the R/V Xuelong cruise over the Ross Sea, Antarctica

Author's Response: Thanks for the reviewer's comment. We have revised it accordingly.

(4) Heading of section 2.2:

Meteorological

Author's Response: Thanks for the reviewer's comment. We have revised it accordingly.

**References**

Bigg, E. K. and Leck, C.: The composition of fragments of bubbles bursting at the ocean surface, Journal of Geophysical Research: Atmospheres, 113https://doi.org/10.1029/2007jd009078, 2008.

Collins, D. B., Zhao, D. F., Ruppel, M. J., Laskina, O., Grandquist, J. R., Modini, R. L., Stokes, M. D., Russell, L. M., Bertram, T. H., Grassian, V. H., Deane, G. B., and Prather, K. A.: Direct aerosol chemical composition measurements to evaluate the physicochemical differences between controlled sea spray aerosol generation schemes, Atmos. Meas. Tech., 7, 3667-3683, https://doi.org/10.5194/amt-7-3667-2014, 2014.

Gaston, C. J., Furutani, H., Guazzotti, S. A., Coffee, K. R., Bates, T. S., Quinn, P. K., Aluwihare, L. I., Mitchell, B. G., and Prather, K. A.: Unique ocean-derived particles serve as a proxy for changes in ocean chemistry, Journal of Geophysical Research: Atmospheres, 116https://doi.org/10.1029/2010jd015289, 2011.

Kirpes, R. M., Bonanno, D., May, N. W., Fraund, M., Barget, A. J., Moffet, R. C., Ault, A. P., and Pratt, K. A.: Wintertime Arctic Sea Spray Aerosol Composition Controlled by Sea Ice Lead Microbiology, Acs Central Sci, 5, 1760-1767, https://doi.org/10.1021/acscentsci.9b00541, 2019.

Krembs, C., Eicken, H., and Deming, J. W.: Exopolymer alteration of physical properties of sea ice and implications for ice habitability and biogeochemistry in a warmer Arctic, P Natl Acad Sci USA, 108, 3653-3658, https://doi.org/10.1073/pnas.1100701108, 2011.

Leck, C. and Bigg, E. K.: Biogenic particles in the surface microlayer and overlaying atmosphere in the central Arctic Ocean during summer, Tellus B, 57, 305-316, https://doi.org/10.1111/j.1600-0889.2005.00148.x, 2005a.

Leck, C. and Bigg, E. K.: Source and evolution of the marine aerosol - A new perspective, Geophys Res Lett, 32https://doi.org/10.1029/2005gl023651, 2005b.

Leck, C. and Bigg, E. K.: New Particle Formation of Marine Biological Origin, Aerosol Sci Tech, 44, 570-577, https://doi.org/10.1080/02786826.2010.481222, 2010.

Leck, C., Gao, Q., Mashayekhy Rad, F., and Nilsson, U.: Size-resolved atmospheric particulate polysaccharides in the high summer Arctic, Atmos Chem Phys, 13, 12573-12588, https://doi.org/10.5194/acp-13-12573-2013, 2013.

Orellana, M. V., Hansell, D. A., Matrai, P. A., and Leck, C.: Marine Polymer-Gels' Relevance in the Atmosphere as Aerosols and CCN, Gels, 7https://doi.org/10.3390/gels7040185, 2021.

Orellana, M. V., Matrai, P. A., Leck, C., Rauschenberg, C. D., Lee, A. M., and Coz, E.: Marine microgels
as a source of cloud condensation nuclei in the high Arctic, P Natl Acad Sci USA, 108, 13612-
13617, https://doi.org/10.1073/pnas.1102457108, 2011.
Verdugo, P.: Marine microgels, Annual Review of Marine Science, 4, 375-400,
https://doi.org/10.1146/annurev-marine-120709-142759, 2012.
Verdugo, P., Alldredge, A. L., Azam, F., Kirchman, D. L., Passow, U., and Santschi, P. H.: The oceanic
gel phase: a bridge in the DOM-POM continuum, Mar Chem, 92, 67-85,
https://doi.org/10.1016/j.marchem.2004.06.017, 2004.

---

## Author Response (AR2)

Dear editor, dear referee,

Thank you for your valuable time to check our manuscript, and we apologize for the errors in the revised manuscript. We have reviewed and revised thoroughly according to your comments.

Sincerely,

Bojiang Su on behalf of all the authors

In this document, the review comments are shown in black. The author's response is shown in blue. The revision is shown in red.

**Responses to Technical comments by Editor**

1) line 94. Please write 34th with "th" in superscript.

Author's Response: Thanks for the technical comment. It has been revised accordingly.

2) line 136. .. the legs I and II.

Author's Response: Thanks for the technical comment. It has been revised accordingly.

3) line 151. Please remove China for the IGAC.

Author's Response: Thanks for the technical comment. It has been revised as suggested. We also removed the countries of all the manufacturer information for consistency.

4) line 208. Please add USA or remove the countries in the all the manufacturer information for consistency.

Author's Response: Thanks for the technical comment. It has been revised as suggested.

5) line 513-514. Please write "th" in superscript.

Author's Response: Thanks for the technical comment. It has been revised accordingly.

**Responses to Technical comments by Referee 2**

- Maps in Fig 5: Please rotate the maps (c) to (l) in Fig 5 by 180 degrees, such that the orientation is similar to the maps in Fig 5(a,b).

Author's Response: Thanks for the reviewer's constructive suggestion. We replotted the Figure 5 as suggested.

[Figure]

Figure 5

- line 38: Antarctic summer

Author's Response: Thanks for the technical comment. It has been revised accordingly.

- line 206: bi-polar, not bio-polar

Author's Response: Thanks for the technical comment. It has been revised accordingly.

- line 252 "the absence of no sea ice" -> "the absence of sea ice"

Author's Response: Thanks for the technical comment. It has been revised accordingly.

- line 282 I suggest "...we set a threshold for the ion count rate of m/z 40..."

Author's Response: Many thanks for the reviewer's constructive suggestion. It has been revised accordingly.

- line 389: "Furthermore, they are likely to be present ..."

Author's Response: Thanks for the technical comment. It has been revised accordingly.